# AwareCompiler: Agentic Context-Aware Compiler Optimization via a Synergistic Knowledge-Data Driven Framework

## Abstract

Compiler optimization is crucial for enhancing program performance by transforming the sequence of optimization passes while maintaining correctness. Despite the promising potential of large language models (LLMs)-based agent for software optimization, automating compiler optimization remains challenging due to: (1) semantic misalignment between abstract program representations and concrete optimization passes, (2) inefficient interaction mechanisms between agents and compiler environments, and (3) reward sparsity from the extensive decision-making process within large optimization spaces. This paper introduces **AwareCompiler**, an agentic framework for compiler optimization that addresses these challenges through three key innovations: structured knowledge integration and dataset construction, knowledge-driven adaptive pass generation, and data-driven hybrid training pipeline. Experimental results on standard benchmarks demonstrate that AwareCompiler significantly outperforms existing baselines in both performance and efficiency, highlighting the effectiveness of our synergistic knowledge-data-driven approach. Our code is publicly available at [1].

## 1 Introduction

Compiler optimization is essential for modern computing systems Aho et al. (2006), involving the automated selection and scheduling of optimization passes from a vast space to enhance program performance (see Figure 1). Among the various optimization objectives, **code size** is a key metric Cummins et al. (2021), as reducing it lowers memory overhead, shortens compilation time, and often improves runtime efficiency. As software systems grow in complexity, the demand for efficient executable code becomes even more critical Gong et al. (2025).

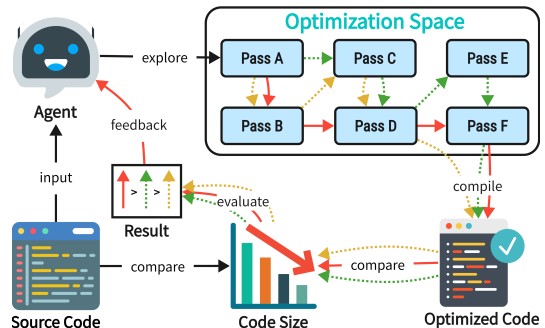

Figure 1: Overview of compiler optimization task.

Historically, compiler optimization relied on handcrafted heuristics and static cost models for pass selection, which were labor-intensive and time-consuming Bergstra et al. (2011); Chen et al. (2012); Ansel et al. (2014). With the rise of data-driven methods, machine learning (ML) has automated pass selection and cost modeling, improving performance but at the cost of high profiling and extensive compilation Haj-Ali et al. (2020); Chen et al. (2021); Zhu et al. (2024). Recently, large language models (LLMs)-based agents have shown promise for compiler optimization Cummins et al. (2024); Deng et al. (2025); Pan et al. (2025a), leveraging pre-trained models to understand and generate code across diverse codebases, even without explicit profiling. This holds potential to overcome the scalability challenges faced by traditional heuristic and ML-based approaches.

However, LLM-based agents often produce ineffective or invalid optimization passes due to insufficient contextual reasoning and an inability to predict the real-world effects of optimizations, leading

---

[1]Code: https://anonymous.4open.science/r/AwareCompiler-4935

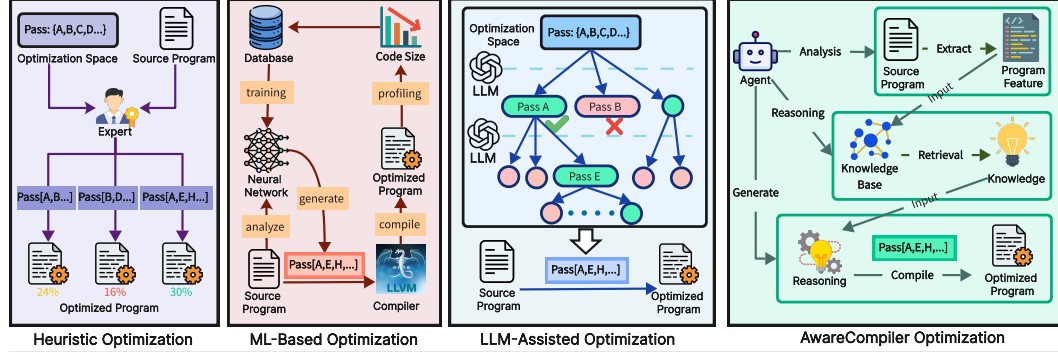

Figure 2: Comparison of different compiler optimization methods, including heuristic-based optimization through the permutations and combinations of "pass", machine learning-based optimization using neural networks, LLM-assisted tuning with hierarchical-based search, and our proposed AwareCompiler method, which incorporates a synergistic knowledge-data driven method.

to performance degradation or even program crashes. Addressing these shortcomings requires tackling three critical challenges: (1) **Semantic misalignment** between abstract program representations and concrete optimization passes, which leads to plausible but incorrect strategies; (2) **Ineffective interaction** mechanisms between agents and compiler environments, which typically rely on brute-force exploration; and (3) **Reward sparsity**, arising from the extensive decision-making horizon in large optimization spaces, preventing effective feedback usage.

This paper introduces **AwareCompiler**, an LLM-based agentic framework designed to address these challenges. AwareCompiler employs a synergistic knowledge-data-driven approach, empowering agents with agentic context awareness to make context-aware optimization sequences generation. Specifically, AwareCompiler contributes three key innovations: **(1) Structured Knowledge Integration and Dataset Construction**: We build a symbolic knowledge base that bridges the semantic gap between program representations and pass-level optimizations, along with a tailored high-quality reasoning dataset. **(2) Knowledge-driven Adaptive Pass Generation**: We propose an adaptive reasoning paradigm that enables agents to analyze code features, retrieve relevant knowledge, and generate optimized sequences, thereby enhancing optimization effectiveness. **(3) Data-driven Hybrid Training Pipeline**: We model compiler optimization as a multi-turn agent-environment interaction problem and optimize the agent through a hybrid training pipeline incorporating an outcome-based composite reward function, ensuring robust and efficient learning.

We evaluate **AwareCompiler** through extensive experiments on standard benchmarks. The results show that AwareCompiler significantly outperforms existing baselines in code size reduction. Beyond performance improvements, AwareCompiler reduces inconsistent or infeasible optimization passes by incorporating internal reasoning and external knowledge based on program context, ensuring more reliable optimizations. By seamlessly combining knowledge-driven reasoning with data-driven learning, AwareCompiler establishes a solid foundation for next-generation LLM-based compiler optimization agents, paving the way for more efficient and powerful compiler architectures.

## 2 PRELIMINARIES

### 2.1 DEFINITION: OPTIMIZATION PASS AND SPACE

The optimization space $\mathcal{P} = \{(p_j, \text{sem}(p_j), \text{deps}(p_j), \text{conf}(p_j))\}$ encodes each pass $p_j$ along with its semantics, dependencies, and conflicts. Here, $\text{sem}(p_j)$ represents the transformation effect of pass $p_j$ on the IR, $\text{deps}(p_j)$ defines the passes that must precede $p_j$, and $\text{conf}(p_j)$ denotes the conflicting passes with $p_j$. An optimization pass sequence is defined as:

$$\pi = (p_1, p_2, \ldots, p_T), \quad p_t \in \mathcal{P}, \quad 1 \le t \le T, \tag{1}$$

where $T$ is the sequence length. Valid pass sequences must satisfy two constraints: (1) dependency relation: if $p_i \in \text{deps}(p_j)$, then $p_i$ precedes $p_j$; and (2) conflict resolution: if $p_i \in \text{conf}(p_j)$, then $p_i$

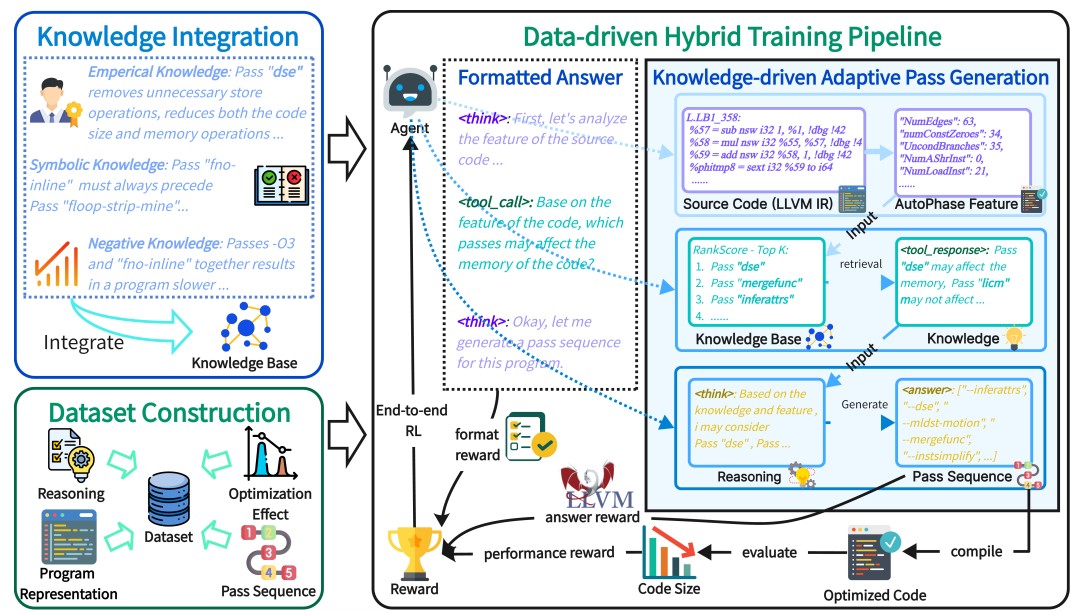

Figure 3: Overview of the framework: AwareCompiler utilizes a synergistic knowledge-data-driven approach to compiler optimization, integrating knowledge-driven adaptive pass generation based on a comprehensive knowledge base (including empirical, symbolic, and negative knowledge) with a data-driven hybrid training pipeline that leverages a curated dataset for model training.

and $p_j$ cannot appear together:

$$\forall p_i, p_j \in \pi, \quad ((p_i \in \mathrm{deps}(p_j)) \Rightarrow \mathrm{pos}(p_i) < \mathrm{pos}(p_j)) \wedge ((p_i \in \mathrm{conf}(p_j)) \Rightarrow \mathrm{pos}(p_i) \neq \mathrm{pos}(p_j)) \tag{2}$$

Let $x \in \mathcal{X}$ represent the source code. Applying sequence $\pi$ to $x$ results in the optimized code $x_{\mathrm{opt}} = \mathcal{C}(x, \pi) \in \mathcal{X}$, where $\mathcal{C}$ is the compiler environment and $\mathcal{X}$ is the code space.

### 2.2 PROBLEM FORMULATION: COMPILER OPTIMIZATION

We formalize compiler optimization for code size reduction as a sequential decision-making problem, where an agent interacts with the compiler environment to iteratively apply passes that transform the program's intermediate representation (IR). The objective of **code size reduction** can be formalized as $\mathcal{L}_{\mathrm{size}}(x, \pi) = \mathrm{Size}(\mathcal{C}(x, \pi))$, where $\mathrm{Size}(\cdot)$ measures the code size. The agent aims to learn a policy $\pi_\theta : \mathcal{S} \to \mathcal{P}$ parameterized by $\theta$, that minimizes the expected code size while maintaining program correctness:

$$\pi_\theta^* = \arg\min_{\pi \in \mathcal{P}} \mathbb{E}_{\pi_\theta} \left[ \mathcal{L}_{\mathrm{size}}(x, \pi_\theta) \right], \tag{3}$$

## 3 METHODOLOGY

In this section, we introduce AwareCompiler, a framework that integrates structured knowledge mapping and dataset construction, data-driven hybrid training pipeline, and knowledge-driven adaptive reasoning paradigm, as illustrated in Figure 3.

### 3.1 STRUCTURED KNOWLEDGE INTEGRATION AND DATASET CONSTRUCTION

AwareCompiler leverages a knowledge-data driven method to combine structured compiler knowledge with large-scale training data. Knowledge-driven strategies utilize domain-specific expertise and formal reasoning while data-driven methods excel at identifying patterns in empirical data.

**Domain-Specific Knowledge Integration.** The knowledge base $\mathcal{K} = \{\mathcal{K}_{\mathrm{emp}}, \mathcal{K}_{\mathrm{sym}}, \mathcal{K}_{\mathrm{neg}}\}$ encodes compiler-specific domain knowledge, enabling flexible retrieval when the agent's internal capabili-

ties are insufficient. It is constructed from three primary sources:

*(i) Empirical Knowledge*: Captures heuristics and patterns learned from historical optimization data, suggesting optimal pass sequences for code features. We represent empirical knowledge as a function $\mathcal{K}_{\text{emp}}$, mapping code features $\mathbf{x}_i$ to optimal pass sequences $\pi_i^*$. From each optimized sequence, we record: the program's AutoPhase feature vector, the optimal pass sequence discovered, and the measured improvement. These produce ¡features, sequence, effect¿ triples that populate the empirical layer of the knowledge base, expressed as:

$$\mathcal{K}_{\text{emp}} : \mathcal{X} \to \mathcal{P}, \quad \mathbf{x}_i \mapsto \pi_i^*, \quad \pi^* = \arg\min_{\pi \in \mathcal{P}} \mathbb{E}_{\mathbf{x}_i \sim \mathcal{X}} \left[ \mathcal{L}_{\text{size}}(x, \pi) \right] \tag{4}$$

*(ii) Symbolic Knowledge*: We gather the semantics, dependencies, and constraints of each optimization pass using opt –help and LLVM's documented transformation rules. This provides structured information such as which passes must precede others, which passes conflict, and what IR transformations each pass performs. These form the symbolic layer of the knowledge base. For each program in the training set, we run a constrained forward search (CFSAT) to obtain locally optimal pass sequences under code-size reduction. From each optimized sequence, we record: Models the structural relationships between optimization passes, ensuring valid sequences by respecting dependency and conflict constraints, which are captured by the function $\text{deps}(p_j) = \{p_i \mid p_i \text{ must precede } p_j\}$ and $\text{conf}(p_j) = \{p_k \mid p_k \text{ conflicts with } p_j\}$:

$$K_{\text{sym}} = \{p_j, \text{deps}(p_j), \text{conf}(p_j)\}, \text{deps}(p_j) = \{p_i \mid p_i \prec p_j\}, \text{conf}(p_j) = \{p_k \mid p_k \perp p_j\} \tag{5}$$

*(iii) Negative Knowledge*: Identifies pass sequences that cause performance regressions or undesired effects indicated by evaluation function $\mathcal{E}$ and threshold $\epsilon$. These sequences are cataloged in $\mathcal{K}$ as *non-optimal sequences*, and the knowledge base prevents their future use. We formalize this as:

$$\mathcal{K}_{\text{neg}} = \{(p_1, p_2, \ldots, p_k) \mid \mathcal{E}(p_1, p_2, \ldots, p_k) < \epsilon\}, \tag{6}$$

Together, these three components form a hybrid symbolic–empirical-negative knowledge base that captures both the structural rules of LLVM passes and the real-world behavior observed across diverse programs.

**Context-Aware Dataset Construction.** To facilitate agent training, we construct a high-quality reasoning dataset $\mathcal{D} = \{(\mathbf{x}_i, \mathcal{T}_i, \pi_i^*, \mathcal{E}_i)\}_{i=1}^N$. *Program representation* $\mathbf{x}_i$ captures the code's statistical features via AutoPhase Haj-Ali et al. (2020), ensuring it remains within the context window limit. *Reasoning process* $\mathcal{T}_i$ encapsulates prompt templates that guide the model through deep reasoning and iterative optimization, fostering contextual awareness. *Pass sequence* $\pi_i^*$ represents the optimal pass sequence derived from expert annotations or learned heuristics, offering guidance during training. *Optimization effect* $\mathcal{E}_i$ quantifies the code size reduction after applying $\pi_i^*$, reflecting the program's efficiency improvement. This context-aware dataset enables the compiler agent to learn the relationships between code features, reasoning steps, pass trajectories, and optimization effects, supporting adaptive reasoning across diverse program context. Our dataset spans a total of 19,603 training programs and 335 test programs, covering both curated and uncurated sources. Specifically, the training set consists of 13,603 uncurated and 6,000 curated programs, while the test set includes 151 uncurated and 184 curated programs. This diverse composition ensures broad coverage of real-world program structures and supports reliable generalization evaluation.

---

**An Example of Training Sample**

`"Program Representation"`: "NumEdges": 63, "numConstZeroes": 34, "Uncond-Branches": 35, "NumAShrInst": 0, ...

`"Reasoning Process"`: Analyzing the autophase features: Total instructions = 1200, Total blocks = 200, Memory instructions = 800, Branch count = 150. Given the high number of memory instructions and blocks, I will prioritize memory and control-flow optimizations. Let me first try an initial pass sequence and check its effectiveness...

`"Pass Sequence"`: ["–inferattrs", "–dse", "–mldst-motion", "–mergefunc", "–instsimplify", "–correlated-propagation", "–slp-vectorizer", "–early-cse-memssa", "–gvn", ...]

`"Optimization Effect"`: "Status": "success", "Improvement (over_oz)": 17.72%.

---

## 3.2 Knowledge-driven Adaptive Pass Generation

AwareCompiler empowers agents with *context awareness* through a knowledge-driven adaptive reasoning paradigm, enabling them to dynamically generate the optimization sequence based on code feature and knowledge assistance, thereby mitigating the brittleness of direct optimization.

**Code Feature Extraction.** The first step involves identifying critical syntactic and semantic features that influence optimization decisions, such as instruction count, control flow complexity, and memory access patterns. Given a source code $x \in \mathcal{X}$, we define a feature extraction function $\mathcal{F}(\cdot)$ that maps $x$ to a higher-dimensional feature space $\mathcal{Z}$ via a nonlinear transformation:

$$\mathbf{z} = \mathcal{F}(x) \in \mathcal{Z}, \quad \mathcal{Z} = \{z_1, z_2, \ldots, z_m\}, \quad z_i = \phi_i(z), \quad \phi_i : \mathcal{Z} \to \mathbb{R}^{d_i}, \quad d_i > 1 \quad (7)$$

The feature vector $\mathbf{z}$ combines high-dimensional statistical and latent semantic features, with each $z_i$ representing an embedding from the feature map $\phi_i$. The extracted $\mathbf{z}$ provides a comprehensive representation of the program's structure, which serves as the foundation for the subsequent domain knowledge retrieval and optimization sequence generation.

**Domain Knowledge Retrieval.** The second step in reasoning involves retrieving relevant knowledge from the compiler knowledge base $\mathcal{K}$ using a rank-based fusion mechanism, RankScore$(\cdot)$, which balances (i) the similarity between the code's feature vector $\mathbf{z}$ and the pass sequence $\pi$, quantified by $\sum_{z_i \in \mathbf{z}} \text{sim}(\phi_i(\mathbf{z}), \phi(\pi))$, and (ii) the minimization of the code size $\mathbb{E}_{\mathbf{z} \sim \mathcal{Z}} [L_{\text{size}}(\mathbf{z}, \pi)]$, guiding the agent to select the most relevant knowledge for optimization:

$$\mathcal{R}(\mathbf{z}, \mathcal{K}) = \text{Top-K} \left( \left\{ \text{RankScore} \left( \sum_{z_i \in \mathbf{z}, \pi \in \mathcal{K}} \text{sim}(\phi_i(\mathbf{z}), \phi(\pi)), \mathbb{E}_{\mathbf{z} \sim \mathcal{Z}} [\mathcal{L}_{\text{size}}(\mathbf{z}, \pi)] \right) \right\} \right) \quad (8)$$

**Pass Sequence Generation.** The final step involves generating an optimal pass sequence $\pi^*$ that minimizes the code size $L_{\text{size}}(x, \pi)$, while satisfying the semantic constraints. Specifically, the indicator function $\mathbb{I}[\cdot]$ ensures the validity of the pass sequence by respecting dependencies and avoiding conflicts. The whole process can be formulated as:

$$\pi^* = \arg \min_{\pi \in \mathcal{R}(\mathbf{z}, \mathcal{K})} \mathbb{E}_\pi [L_{\text{size}}(x, \pi)] \text{ subject to } \sum_{p_i, p_j \in \pi} \mathbb{I}[p_i \in \text{deps}(p_j)] \quad \text{and} \quad \mathbb{I}[p_i \in \text{conf}(p_j)] \quad (9)$$

### 3.3 Data-driven Hybrid Training Pipeline

To integrate knowledge and data effectively, AwareCompiler adopts a two-stage training approach. The first stage, Supervised Fine-Tuning (SFT), trains the model to follow specific reasoning format and identify effective pass sequences quickly. The second stage introduces Reinforcement Learning (RL), where a composite reward function balances performance gains with penalties for violations, evolving the agent from accuracy to effectiveness.

**Supervised Fine-Tuning.** During the SFT phase, the agent learns optimization heuristics to solve various tasks. This enables the agent to internalize fundamental optimization patterns, follow specific reasoning formats, invoke knowledge base retrieval, and generate valid sequences. The SFT objective is to align the agent's policy $\hat{\pi}^{SFT}$ with expert-defined strategies $\hat{\pi}^{SFT} = \arg \min_{\pi \in \mathcal{P}} \mathcal{L}_{size}(\pi, \mathbf{x}, \mathcal{D})$.

**Reinforcement Learning.** The RL phase refines the agent's decision-making by exploring optimization paths and receiving rewards that balance performance with constraint penalties. The goal is to learn a policy $\hat{\pi}^{RL}$ that maximizes long-term cumulative rewards:

$$\hat{\pi}^{RL} = \arg \max_\pi \mathbb{E}_\pi \left[ \sum_{t=0}^{T} \gamma^t r_t \right] \quad (10)$$

where $r_t$ is the reward at time $t$, and $\gamma$ is the discount factor. The composite reward function aligns outputs with format standardization, answer validity, and performance improvement:

- *Format Reward*: Encourage the model follows structured decision-making: reasoning within `<think>`, retrieve knowledge base within `<tool_call>`, and generate pass sequence within `<answer>`. A score of 1 point is awarded for valid format and 0 otherwise.

- *Answer Reward*: Validates the answer through a compilation test, with schema constraints and tool protocols are checked respectively, ensuring the generated optimization passes are syntactically and structurally correct.

- *Performance Reward*: Measures the reduction in code size $\Delta IC = \{IC_{\text{before}} - IC_{\text{after}}\}/IC_{\text{before}}$, as $IC_{\text{before}}, IC_{\text{after}}$ denote the instruction counts before and after applying the candidate sequence, respectively.

**End-to-End Training and Synergy.** The two-stage, synergistic training process, consisting of SFT followed by RL, combines stable learning with dynamic adaptation, enabling AwareCompiler to progressively learn from simple heuristics to optimized solutions. The joint objective of the whole learning process can be described as:

$$\pi^{\text{final}} = \arg\max_{\pi} \left( -\mathcal{L}_{size}(\pi, \mathbf{x}, \mathcal{D}_{SFT}) + \mathbb{E}_{\pi} \left[ \sum_{t=0}^{T} \gamma^t r_t \right] \right) \quad (11)$$

This synergy ensures the agent satisfies constraints while significantly improving performance.

Table 1: Comparison of code size reduction across heuristic, ML-based, LLM-assisted, and our proposed AwareCompiler on various benchmarks. Results are reported as percentage reductions in LLVM IR count, where higher values indicate greater effectiveness.

| Method | blas | cbench | chstone | mibench | npb | opencv | tensorflow | Avg. |
|---|---|---|---|---|---|---|---|---|
| *Heuristic Optimization* | | | | | | | | |
| Opt -O1 | 4.78% | 50.87% | 53.10% | 59.05% | 36.52% | 3.11% | 1.74% | 29.88% |
| Opt -O2 | 5.88% | 27.23% | 31.98% | 44.39% | 34.26% | 2.00% | 1.25% | 21.00% |
| Opt -O3 | 4.64% | 15.96% | 28.87% | 21.46% | 27.93% | -8.90% | -0.77% | 12.74% |
| Opt -Oz | 5.70% | 51.93% | 50.56% | 58.29% | 38.21% | 3.38% | 1.77% | 29.98% |
| *ML-based Optimization* | | | | | | | | |
| CompTuner | 5.25% | 49.01% | 50.13% | 55.09% | 44.64% | 3.22% | 1.70% | 29.86% |
| BOCA | 5.29% | 49.26% | 49.56% | 55.00% | 44.60% | 3.25% | 1.70% | 29.81% |
| AutoPhase | 5.43% | 49.45% | 48.15% | 55.51% | 36.39% | 3.30% | 2.60% | 30.03% |
| *LLM-assisted Optimization* | | | | | | | | |
| GPT-5 | 4.35% | 25.69% | 11.24% | 27.18% | 19.70% | 0.34% | 1.90% | 12.91% |
| Gemini-2.5-pro | 1.21% | 17.93% | 25.14% | 39.61% | 35.91% | -0.36% | -0.26% | 17.03% |
| DeepSeek-V3 | 1.05% | 49.05% | 48.55% | 57.62% | 31.21% | -1.57% | -0.28% | 26.52% |
| Claude-opus-4 | 2.58% | 47.44% | 48.87% | 58.65% | 0.00% | 1.51% | 0.25% | 22.76% |
| GLM-4.5 | 2.41% | 39.79% | 42.93% | 48.72% | 33.81% | 1.05% | 0.20% | 24.13% |
| Hunyuan-A13B-Instruct | 0.67% | 15.55% | 28.93% | 25.03% | 14.78% | 1.08% | 0.62% | 12.38% |
| Kimi-Dev-72B | 1.03% | 46.87% | 47.67% | 54.71% | 34.20% | 2.87% | 1.39% | 26.96% |
| Qwen3-235B-A22B | 0.56% | 30.60% | 21.29% | 19.23% | 14.51% | 0.99% | 0.78% | 12.57% |
| Qwen3-Coder-480B-A35B | 1.47% | 37.04% | 36.89% | 43.16% | 23.38% | -3.78% | 0.68% | 19.83% |
| QwenLong-L1-32B | 0.06% | 13.04% | 16.73% | 16.73% | 12.14% | 0.60% | 0.17% | 8.50% |
| *Ours* | | | | | | | | |
| **AwareCompiler-1.5B** | 5.45% | 51.93% | 49.91% | 58.29% | 38.73% | 3.30% | 2.60% | 30.03% |
| **AwareCompiler-3B** | 6.32% | 47.67% | 45.22% | 49.97% | 40.09% | 4.47% | 1.29% | 27.86% |
| **AwareCompiler-7B** | 4.94% | 50.55% | 49.44% | 57.04% | 39.36% | 2.88% | 2.06% | 29.47% |

## 4 EXPERIMENT

We evaluate the effectiveness of **AwareCompiler** through a series of empirical studies aimed at addressing the following research questions (RQs): **RQ1:** How does AwareCompiler perform in comparison to existing baselines in compiler optimization? **RQ2:** Does knowledge-driven reasoning in AwareCompiler effectively address the semantic misalignment challenge? **RQ3:** What is the impact of the various driving methods (data and knowledge) employed in AwareCompiler? **RQ4:** How do the different reward functions in AwareCompiler contribute to its optimization performance? **RQ5:** What are the key details of AwareCompiler's reasoning and optimization process?

## 4.1 Experimental Setup

**Benchmarks.** We evaluate AwareCompiler on a diverse collection of benchmark suites: blas Dongarra et al. (1990), npb Bailey et al. (1995), cbench Chen & Karp (1997), opencv Bradski (2000), mibench Guthaus et al. (2001), chstone Miyamoto et al. (2010), tensorflow Abadi et al. (2016), providing a comprehensive evaluation across domains.

**Baselines.** We compare AwareCompiler against several compiler optimization methods, including: (1) **Heuristic Methods**: expert optimization based on empirical knowledge, represented by `opt -O1`, `opt -O2`, `opt -O3`, and `opt -Oz` Krentel & Ewing (1990); (2) **Machine Learning (ML) Methods**: ML-driven optimization approaches, including RL-based, Bayesian, and heuristic-search hybrids, such as AutoPhase Haj-Ali et al. (2020), BOCA Chen et al. (2021), and CompTuner Zhu et al. (2024); and (3) **Vanilla LLMs**: mainstream LLMs using few-shot prompting with chain-of-thought, such as GPT-5 OpenAI (2025), DeepSeek-V3 et al. (2025a), and Gemini-2.5 et al. (2025b).

**Training Details.** AwareCompiler was trained using the Qwen2.5-instruct models (1.5B, 3B, 7B) Qwen et al. (2025). The training pipeline includes 2,000+ SFT samples for policy initialization, followed by RL on 15,000+ samples for policy enhancement. Experiments were conducted on Intel Xeon Gold 6430 servers (128 cores, 1TB RAM) with NVIDIA H100 GPUs (4×80GB HBM3).

**Evaluation Details.** We use LLVM Lattner & Adve (2004) IR count as the representation of code size, and average percentage of IR instruction reduction as the metric of optimization. All baseline models operate within a fixed optimization space consisting of 124 LLVM 10.0.0 optimization passes, ensuring a fair comparison of performance across different methods.

## 4.2 Main Results (RQ1)

**Experiment Objective.** This experiment evaluates AwareCompiler's effectiveness in reducing code size across various benchmarks, using LLVM IR count as the optimization metric, where higher values indicate better performance.

**Result Analysis.** As shown in Table 1, AwareCompiler consistently outperforms LLM-assisted methods, achieving reductions comparable to expert-level optimizations. Compared to advanced models like GPT-5 (12.91%) and DeepSeek-V3 (26.52%), AwareCompiler demonstrates significant improvements, even with smaller models. This highlights the effectiveness of its knowledge-data-driven approach, where knowledge-driven reasoning and data-driven optimization are seamlessly integrated. The key advantage of AwareCompiler lies in its ability to perform dynamic, context-aware optimizations, generating valid and effective pass sequences for complex tasks while reducing reliance on brute-force methods typically employed by LLMs.

Additionally, we have expanded our evaluation to include three representative ML baselines. The table above summarizes the evaluation results, showing the optimization performance (code size reduction rate) and the optimization time required for each method. By analyzing the data in the table, we observed two key findings:

- Optimization Performance: AwareCompiler consistently achieves competitive or superior optimization quality across the seven benchmark suites. Its average code-size reduction of 30.03% slightly surpasses the best-performing ML baseline, CompTuner (29.86%). This demonstrates that our agentic framework—combining symbolic knowledge with data-driven reasoning—provides more effective and stable optimization compared to traditional ML-based methods.
- Optimization Speed: AwareCompiler offers a significant advantage in speed. While CompTuner requires 9800 seconds/program, and BOCA takes 2700 seconds/program, AwareCompiler optimizes the code in 8 seconds/program, which translates to a 340×–1200× speedup over the best-performing ML baselines. This efficiency is achieved without compromising the quality of optimization, highlighting AwareCompiler's ability to generate optimized passes quickly and effectively.

**Summary.** AwareCompiler's ability to achieve expert-level optimization with its unique knowledge-data-driven framework across various benchmarks, not only positions it as an effective and reli-

able solution for compiler optimization, but also demonstrates the potential of integrating structured knowledge with learning-based systems to overcome the scalability challenges faced by traditional optimization methods.

## 4.3 SEMANTIC MISALIGNMENT ANALYSIS (RQ2)

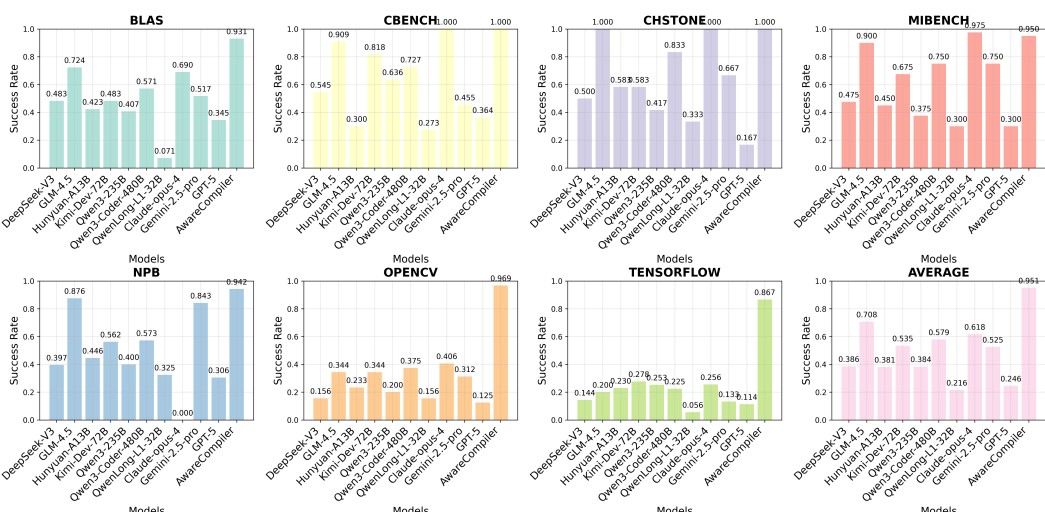

Figure 4: Comparison of success rates in generating valid pass sequences across different models.

**Experiment Objective.** We evaluate the effectiveness of the AwareCompiler by measuring its success rate across various benchmarks. The success rate quantifies the proportion of valid optimization passes generated, which directly impacts the quality and efficiency of compiler optimization.

**Result Analysis.** As shown in Figure 4, AwareCompiler achieves the highest success rates across benchmarks, with near-perfect results in CBench and CHSTONE. This highlights the effectiveness of its knowledge-driven adaptive pass generation in mitigating the **semantic misalignment challenge**. In contrast, LLMs like GPT-5, Gemini-2.5-pro, and Claude-opus-4 show lower success rates, especially in OpenCV and TensorFlow, reflecting their limitations in building accurate semantic links between program representations and optimization passes.

**Summary.** The superior success rates of AwareCompiler demonstrate the effectiveness of its integrated knowledge base and knowledge-driven adaptive pass generation mechanism, which bridges the semantic gap between abstract program representations and concrete optimization passes, significantly enhancing the efficiency, reliability, and robustness of compiler optimizations.

## 4.4 ABLATION STUDY (RQ3)

**Experiment Objective.** We conducted an ablation study to assess the contributions of key components in AwareCompiler: the knowledge-driven adaptive reasoning ("w/o knowledge" by removing the knowledge base), the data-driven training pipeline ("w/o data" by using only SFT), and their combined effect ("w/o knowledge & data" by using the base model).

**Result Analysis.** As shown in Table 2, "w/o knowledge" leads to a noticeable performance drop, emphasizing the importance of context-aware knowledge retrieval. The "w/o data" exhibits a similar decrease, highlighting the value of the RL training. The "w/o knowledge & data" shows the largest reduction, underscoring the necessity of both components for effective optimization.

As for why removing data does not significantly reduce accuracy for AwareCompiler-7B on the BLAS dataset, we attribute this to two factors.

- **Structural homogeneity of BLAS programs**. BLAS kernels exhibit highly regular loop-nest patterns and predictable IR transformations. Effective optimization sequences are al-

Table 2: Ablation study for AwareCompiler, comparing optimization performance across various configurations: with full knowledge and data, and without knowledge, data, or both.

| Method | blas | cbench | chstone | mibench | npb | opencv | tensorflow | Avg. |
|---|---|---|---|---|---|---|---|---|
| **AwareCompiler-1.5B** | 5.45% | 51.93% | 49.91% | 58.29% | 38.73% | 3.30% | 2.60% | 30.03% |
| w/o knowledge | 5.32% | 46.20% | 46.60% | 54.50% | 34.1% | 2.60% | 1.35% | 27.24% |
| w/o data | 4.87% | 51.92% | 50.56% | 55.98% | 37.77% | 2.82% | 1.69% | 29.37% |
| w/o knowledge & data | 0.96% | 36.68% | 24.59% | 33.68% | 21.37% | 0.67% | 0.93% | 16.98% |
| **AwareCompiler-3B** | 6.32% | 47.67% | 45.22% | 49.97% | 40.09% | 4.47% | 1.29% | 27.86% |
| w/o knowledge | 0.66% | 38.60% | 28.54% | 35.77% | 38.83% | 3.18% | 0.81% | 20.91% |
| w/o data | 5.25% | 46.95% | 40.00% | 48.92% | 36.49% | 2.47% | 1.09% | 25.88% |
| w/o knowledge & data | 0.21% | 3.83% | 16.95% | 12.14% | 5.09% | -0.45% | 0.14% | 5.42% |
| **AwareCompiler-7B** | 4.94% | 50.55% | 49.44% | 57.04% | 39.36% | 2.88% | 2.06% | 29.47% |
| w/o knowledge | 1.79% | 50.43% | 48.73% | 54.78% | 40.10% | 2.39% | 1.24% | 28.49% |
| w/o data | 5.42% | 50.43% | 46.34% | 57.01% | 38.01% | 3.03% | 1.68% | 28.82% |
| w/o knowledge & data | 0.96% | 36.58% | 24.59% | 33.68% | 21.37% | 0.67% | 0.93% | 16.97% |

ready well captured in our symbolic and empirical knowledge bases, allowing the model to rely primarily on knowledge rather than data-driven exploration.

- **Stronger reasoning capabilities of the 7B model**. The larger model can solve optimization tasks in such simple, regular domains with minimal reliance on data.

Importantly, this phenomenon is **localized to BLAS**. For more complex benchmarks (e.g., cbench, mibench, npb), the full model—with both knowledge and data—consistently outperforms the "w/o data" variant, underscoring the essential role of data in handling diverse and heterogeneous IR structures.

**Summary.** These results confirm that AwareCompiler's synergistic knowledge-data-driven method, which combines knowledge-driven reasoning with data-driven learning, effectively overcomes the **ineffective interaction challenge**, resulting in superior pass sequences with improved quality.

## 4.5 REWARD SPARSITY ANALYSIS (RQ4)

| Reward Scheme | Performance Gain |
|---|---|
| w/o $R_{format}$ | 8.02% |
| w/o $R_{performance}$ | 12.50% |
| w/o $R_{answer}$ | 18.10% |
| **Full Reward** | 30.03% |

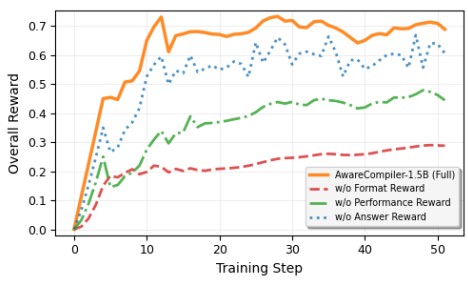

(a) Impact of reward schemes on optimization.   (b) Impact of reward schemes on training.

Figure 5: Impact of reward functions on optimization performance and training efficiency.

**Experiment Objective.** This experiment evaluates the effectiveness of AwareCompiler in addressing the reward sparsity challenge by examining the impact of different reward function configurations on optimization performance and training efficiency.

**Result Analysis.** As shown in Figure 5, the full reward configuration achieves a performance gain of 30.03%. Removing $R_{format}$, $R_{answer}$, or $R_{performance}$ results in performance degradation. Furthermore, the full reward configuration leads to faster model convergence and a higher cumulative reward.

**Summary.** AwareCompiler effectively addresses the **reward sparsity challenge** by utilizing a comprehensive reward function that balances format standardization, answer validity, and performance improvement, thereby ensuring both effective optimizations and efficient learning.

### 4.6 Case Study: Detailed Demonstration of AwareCompiler (RQ5)

---

**A case study of AwareCompiler's agentic workflow**

\<think\>: "Based on autophase features, I generate an initial optimization sequence to reduce the program's instruction count. I will begin with a common heuristic approach."

\<tool_call\>: "Let me verify the performance of the initial heuristic sequence by using the 'instrcount' tool to compare it against the baseline optimization ('-Oz')."

\<think\>: "The initial optimization attempt has resulted in no performance improvement, indicating the need for a more tailored approach."

\<think\>: "Both heuristic attempts have failed. It's clear that the generalized strategies are insufficient for this particular program. I will query the knowledge base for expert advice to generate a more effective pass sequence."

\<tool_response\>: "The knowledge base suggests using the '–gvn' pass to eliminate redundant computations, which has been empirically validated for similar programs."

\<think\>: "I'll integrate the recommended pass into the sequence. This sequence should meet the optimization goal based on the program's characteristics."

\<answer\>: ["–loop-sink", "–gvn", "–ipsccp", ...]

---

**Experiment Objective.** This case study evaluates AwareCompiler's ability to improve performance when heuristics fail and whether the knowledge base can guide the agent to achieve better results.

**Result Analysis.** The agent initially generated an optimization sequence based on autophase features with no improvement. After consulting the knowledge base, the agent applied a tailored sequence, resulting in a 3.2% improvement, highlighting the effectiveness of the knowledge base.

**Summary.** This case study underscores how knowledge-driven adaptive pass generation enables AwareCompiler to overcome the limitations of heuristics and achieve consistent performance gains.

## 5 Related Work

**Compiler Optimization.** Compiler optimization can be classified into three categories: Heuristic optimizations rely on predefined rules and search-based methods to enhance performance Chen et al. (2012); Garciarena & Santana (2016). Machine learning-based methods use models trained on domain-specific datasets to generate optimization sequences, but face scalability challenges in larger, more diverse codebases Haj-Ali et al. (2020); Chen et al. (2021); Zhu et al. (2024); Deng et al. (2025); Pan et al. (2025b). LLM-assisted methods enable agents to autonomously explore optimizations via specialized training pipelines Cummins et al. (2023); Gong et al. (2025). AwareCompiler introduces a synergistic knowledge-data-driven approach that dynamically generates optimization passes, overcoming the limitations of rigid rule-based or static learning methods.

**Agentic Language Models.** Recent advancements in LLMs have led to agentic language models capable of autonomous reasoning, planning, and tool interaction Huang et al. (2024); Zhao et al. (2024). These language agents generate hypotheses and interact with external environments to achieve goals Mei et al. (2024); Hong et al. (2024). A key challenge is grounding their reasoning in specific domains to ensure effective and valid actions Shang et al. (2024); Pan et al. (2025a). AwareCompiler applies this agentic paradigm to compiler optimization, navigating the structured, vast optimization space.

## 6 Conclusion

We introduce **AwareCompiler**, an agentic framework for compiler optimization that leverages a synergistic knowledge-data-driven approach. By integrating knowledge-driven reasoning with data-driven learning, AwareCompiler effectively addresses challenges such as semantic misalignment, inefficient agent-environment interactions, and reward sparsity. Extensive experiments across multiple benchmarks demonstrate that AwareCompiler outperforms existing methods, achieving significant code size reductions while ensuring the validity of optimizations. AwareCompiler lays a robust foundation for more efficient, flexible, and automated compiler optimization in the future.

## ETHICS STATEMENT

AwareCompiler leverages LLMs and domain-specific knowledge and data to automate compiler optimization. While this framework enhances efficiency, ethical concerns must be considered. In particular, the potential for producing suboptimal or invalid optimization passes could destabilize system performance. AwareCompiler mitigates this risk through its knowledge-driven reasoning mechanism and continuous validation processes, but ongoing monitoring and verification are necessary. The framework's reliance on large-scale datasets and LLMs also raises questions regarding data quality and model biases, which must be addressed through responsible usage, transparent methodologies, and regular updates to the underlying models. Ethical deployment requires careful attention to the accuracy, reliability, and potential environmental impact of optimizations generated by AwareCompiler.

## REPRODUCIBILITY

To ensure the reproducibility of AwareCompiler, we provide full access to the source code and experimental setup. All models, training data, and scripts are publicly available at the following URL: https://anonymous.4open.science/r/AwareCompiler-4935. This repository includes detailed instructions for setting up the required environment, performing the experiments, and replicating the results across standard benchmarks. By sharing these resources, we aim to facilitate the verification and replication of our results, supporting further advancements in the field of compiler optimization.

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

APPENDIX

## A PROMPT TEMPLATES

```
Act as a compiler optimization expert to find an optimal LLVM pass sequence that minimizes
    total instruction count.

LLVM IR features:
{formatted_features}
Initial instruction count: {TotalInsts}

Workflow:
1. <think>: Analyze features and reason about effective passes.
2. <tool_call>: Query lightrag_compiler_optimization for a recommended sequence.
3. <tool_call>: Verify the sequence using instrcount with {program_id}.
4. Interpret instrcount result: positive improvement_over_oz means the optimization is
    successful, negative means it is not. Adjust your strategy accordingly.
5. Output only your final pass sequence in <answer> tags as a JSON list.

Example output format:
<|im_start|>assistant
<think> Analyzing the autophase features, I notice a high number of memory instructions and
    branches. I will prioritize memory and control-flow optimizations.
</think>
<tool_call>
{{"name": "lightrag_compiler_optimization", "arguments": {{"query": "{formatted_features}"}}}}
</tool_call>
<|im_end|>

<|im_start|>user <tool_response> {{"recommended_pass_sequence": ["--inferattrs", "--dse",
    "--mldst-motion", "--mergefunc", ...], "performance_improvement": 0.42}}
</tool_response>
<|im_end|>

<|im_start|>assistant
<think> I will verify the recommended sequence using the instrcount tool.
</think>
<tool_call>
{{"name": "instrcount", "arguments": {{"filename": "{program_id}", "optimization_flags":
    ["--inferattrs", "--dse", "--mldst-motion", "--mergefunc", ...]}}}} </tool_call>
<|im_end|>

<|im_start|>user
<tool_response> {{"status": "success", "improvement_over_oz": 0.42}}
</tool_response>
<|im_end|>

<|im_start|>assistant
<answer>
["--inferattrs", "--dse", "--mldst-motion", "--mergefunc", ...]
</answer>
<|im_end|>
```

## B AUTOPHASE FEATURE SET

As referenced in **Feature Extraction and Representation**, our framework utilizes the 56 statistical features from AutoPhase Haj-Ali et al. (2020) to represent programs compactly for the LLM. These features capture various aspects of the program's static structure and instruction mix. Table 3 provides a complete list of these features.

Table 3: List of 56 AutoPhase Features Utilized.

| Index | Feature Description | Index | Feature Description |
|-------|---------------------|-------|---------------------|
| 0 | BBs: total phi args >5 | 28 | Number of And insts |
| 1 | BBs: total phi args in [1,5] | 29 | BBs: instruction count in [15,500] |
| 2 | BBs: count with 1 predecessor | 30 | BBs: instruction count <15 |
| 3 | BBs: count with 1 predecessor and 1 successor | 31 | Number of BitCast insts |
| 4 | BBs: count with 1 predecessor and 2 successors | 32 | Number of Br insts |
| 5 | BBs: count with 1 successor | 33 | Number of Call insts |

Table 3 – continued from previous page

| Index | Feature Description | Index | Feature Description |
|-------|--------------------|-------|--------------------|
| 6 | BBs: count with 2 predecessors | 34 | Number of GetElementPtr insts |
| 7 | BBs: count with 2 predecessors and 1 successor | 35 | Number of ICmp insts |
| 8 | BBs: count with 2 predecessors and successors | 36 | Number of LShr insts |
| 9 | BBs: count with 2 successors | 37 | Number of Load insts |
| 10 | BBs: count with >2 predecessors | 38 | Number of Mul insts |
| 11 | BBs: Phi node count in range (0,3] per BB | 39 | Number of Or insts |
| 12 | BBs: count with more than 3 Phi nodes | 40 | Number of PHI insts |
| 13 | BBs: count with no Phi nodes | 41 | Number of Ret insts |
| 14 | Number of Phi-nodes at beginning of BB | 42 | Number of SExt insts |
| 15 | Number of branches | 43 | Number of Select insts |
| 16 | Number of calls that return an int | 44 | Number of Shl insts |
| 17 | Number of critical edges | 45 | Number of Store insts |
| 18 | Number of edges | 46 | Number of Sub insts |
| 19 | Occurrences of 32-bit integer constants | 47 | Number of Trunc insts |
| 20 | Occurrences of 64-bit integer constants | 48 | Number of Xor insts |
| 21 | Occurrences of constant 0 | 49 | Number of ZExt insts |
| 22 | Occurrences of constant 1 | 50 | Number of basic blocks |
| 23 | Number of unconditional branches | 51 | Number of instructions (all types) |
| 24 | Binary operations with a constant operand | 52 | Number of memory instructions |
| 25 | Number of AShr insts | 53 | Number of non-external functions |
| 26 | Number of Add insts | 54 | Total arguments to Phi nodes |
| 27 | Number of Alloca insts | 55 | Number of Unary operations |

## C    LLVM Optimization Passes

Our AwareCompiler framework and baseline models operate within an action space comprising 124 individual LLVM 10.0.0 `opt` transformation passes, resulting in a total of 125 distinct actions available to the optimization agent. Table 4 enumerates these passes and the `-Oz` along with their corresponding indices used within our system. These flags can typically be obtained from the CompilerGym LLVM environment.

Table 4: List of Utilized LLVM Compiler Passes and `-Oz` with Corresponding Indices.

| Index | Flag | Index | Flag | Index | Flag |
|---|---|---|---|---|---|
| 0 | add-discriminators | 42 | globalsplit | 84 | lower-expect |
| 1 | adce | 43 | guard-widening | 85 | lower-guard-intrinsic |
| 2 | aggressive-instcombine | 44 | hotcoldsplit | 86 | lowerinvoke |
| 3 | alignment-from-assumptions | 45 | ipconstprop | 87 | lower-matrix-intrinsics |
| 4 | always-inline | 46 | ipsccp | 88 | lowerswitch |
| 5 | argpromotion | 47 | indvars | 89 | lower-widenable-condition |
| 6 | attributor | 48 | irce | 90 | memcpyopt |
| 7 | barrier | 49 | infer-address-spaces | 91 | mergefunc |
| 8 | bdce | 50 | inferattrs | 92 | mergeicmps |
| 9 | break-crit-edges | 51 | inject-tli-mappings | 93 | mldst-motion |
| 10 | simplifycfg | 52 | instsimplify | 94 | sancov |
| 11 | callsite-splitting | 53 | instcombine | 95 | name-anon-globals |
| 12 | called-value-propagation | 54 | instnamer | 96 | nary-reassociate |
| 13 | canonicalize-aliases | 55 | jump-threading | 97 | newgvn |
| 14 | consthoist | 56 | lcssa | 98 | pgo-memop-opt |
| 15 | constmerge | 57 | licm | 99 | partial-inliner |
| 16 | constprop | 58 | libcalls-shrinkwrap | 100 | partially-inline-libcalls |
| 17 | coro-cleanup | 59 | load-store-vectorizer | 101 | post-inline-ee-instrument |
| 18 | coro-early | 60 | loop-data-prefetch | 102 | functionattrs |
| 19 | coro-elide | 61 | loop-deletion | 103 | mem2reg |
| 20 | coro-split | 62 | loop-distribute | 104 | prune-eh |
| 21 | correlated-propagation | 63 | loop-fusion | 105 | reassociate |
| 22 | cross-dso-cfi | 64 | loop-guard-widening | 106 | redundant-dbg-inst-elim |
| 23 | deadargelim | 65 | loop-idiom | 107 | rpo-functionattrs |
| 24 | dce | 66 | loop-instsimplify | 108 | rewrite-statepoints-for-gc |
| 25 | die | 67 | loop-interchange | 109 | sccp |
| 26 | dse | 68 | loop-load-elim | 110 | slp-vectorizer |
| 27 | reg2mem | 69 | loop-predication | 111 | sroa |
| 28 | div-rem-pairs | 70 | loop-reroll | 112 | scalarizer |
| 29 | early-cse-memssa | 71 | loop-rotate | 113 | separate-const-offset-from-gep |
| 30 | early-cse | 72 | loop-simplifycfg | 114 | simple-loop-unswitch |
| 31 | elim-avail-extern | 73 | loop-simplify | 115 | sink |
| 32 | ee-instrument | 74 | loop-sink | 116 | speculative-execution |
| 33 | flattencfg | 75 | loop-reduce | 117 | slsr |
| 34 | float2int | 76 | loop-unroll-and-jam | 118 | strip-dead-prototypes |
| 35 | forceattrs | 77 | loop-unroll | 119 | strip-debug-declare |
| 36 | inline | 78 | loop-unswitch | 120 | strip-nondebug |
| 37 | insert-gcov-profiling | 79 | loop-vectorize | 121 | strip |
| 38 | gvn-hoist | 80 | loop-versioning-licm | 122 | tailcallelim |
| 39 | gvn | 81 | loop-versioning | 123 | mergereturn |
| 40 | globaldce | 82 | loweratomic | 124 | -Oz |
| 41 | globalopt | 83 | lower-constant-intrinsics | | |

## D    The Use of Large Language Models

We used LLMs to assist in refining the clarity and coherence of the writing in the paper. The LLMs were specifically employed to improve phrasing, ensure academic rigor, and enhance overall readability. Their contribution was strictly in the writing process, and all content was thoroughly reviewed and finalized by the authors.

