# OpenReview forum: "AwareCompiler: Agentic Context-Aware Compiler Optimization via a Synergistic Knowledge-Data Driven Framework"
_ICLR.cc/2026/Conference — ICLR 2026 Conference Desk Rejected Submission_

### Official Review · Reviewer_ZsmH · 2025-10-29

**Soundness:** 3
**Presentation:** 2
**Contribution:** 2
**Rating:** 4
**Confidence:** 3

**Summary:**

This paper proposes a novel training pipeline of LLM-based compiler, targeting at solving an important problem of reducing generated code size, with the action space of determining the optimization pass sequences within the compiler.
The core contribution of this training pipeline is allowing the model to integrate human-defined domain knowledge dataset in the reasoning process. The model is trained on two stages, one SFT followed by RL.
Experiments shows the trained models used on several compilation tasks reduces the generated code size to the human-expert level (-Oz). In some cases, it is even slightly better than human experts. Moreover, it also shows a high success rate compared to base models.

**Strengths:**

- Solid problem definition with a reasonable solution.
- Strong empirical results. LLM-based methods achieves near human-expert results, sometimes even better
- The evaluations are abundant, covering most essential parts like success rates, ablations and reward design

**Weaknesses:**

- The major goal is reducing the code size, instead of the primary target of compiler optimization, which is producing code that runs faster. Though the author mentions reducing code size **often** improves runtime efficiency, it is not always the case (e.g. -O2 runs faster than -O1, but with bigger code size, from Table 1).
- Building on the previous point, it remains unclear how AwareCompiler affects runtime efficiency, even when code size reduction is the main optimization target. A more explicit discussion or evaluation of this trade-off would strengthen the work.
- Generalizability: Although the results show a slight improvement over -Oz, which is manually designed, there is some concern that the trained model may primarily be distilling knowledge from existing human-designed compilers, particularly during the supervised fine-tuning (SFT) stage. It would enhance the paper’s contribution to provide evidence of transferability, such as demonstrating effectiveness on other compilers or showing that the model can adapt to new tasks with minimal human intervention.

**Questions:**

- Could the authors elaborate on how the context-aware dataset is generated and how it is incorporated into the training process?
- What level of effort is required to generate the training data? Specifically, would it be necessary to recreate the dataset from scratch—potentially with significant effort—when adapting the approach to new tasks or compilers?
- What is the 'data' in 'w/o data' Table 2?
- How does the proposed method affect the runtime performance of the generated code?
- From a theoretical standpoint, could the same training pipeline be applied to optimize other compiler performance metrics, such as runtime speed, instead of code size?

---

> ### Author Response · Authors · 2025-11-21
> **Response to Reviewer ZsmH (Part 1/3)**
>
> Thank you very much for your time and effort in reviewing our paper. We sincerely appreciate your feedback. Below, we respectfully provide our detailed responses to address your concerns.
>
> ---
>
> ### **W1 & W2 & Q4:  How does the proposed method affect the runtime performance of the generated code?**
>
> > The major goal is reducing the code size, instead of the primary target of compiler optimization, which is producing code that runs faster. Though the author mentions reducing code size **often** improves runtime efficiency, it is not always the case (e.g. -O2 runs faster than -O1, but with bigger code size, from Table 1).
> >
>
> > Building on the previous point, it remains unclear how AwareCompiler affects runtime efficiency, even when code size reduction is the main optimization target. A more explicit discussion or evaluation of this trade-off would strengthen the work.
> >
>
> We appreciate the reviewer’s concern that code-size reduction does not always correlate with runtime improvement. To address this, we conducted **additional experiments targeting performance-oriented optimization** rather than size alone.
>
> - Instead of wall-clock execution time — which is highly sensitive to machine noise — we adopt **LLVM-MCA** to estimate the *cycles* of the generated assembly. This approach provides a **stable, hardware-agnostic, and reproducible** performance signal, and is widely used in recent compiler-optimization studies.
> - We evaluate our method on five benchmark suites and report the **average improvement over -O3 in estimated cycles**:
>
> | Benchmark | Average Runtime Performance Over -O3 |
> | --- | --- |
> | cbench | 0.231275 |
> | mibench | 0.130990 |
> | blas | 0.029383 |
> | opencv | 0.072103 |
> | chstone | 0.176963 |
> - These results show that our approach provides a **balanced trade-off between code size and execution performance**. The model does not merely shrink code; it also discovers optimization sequences that **outperform -O3 in estimated runtime**, demonstrating that the learned policy is not biased toward code-size reduction alone but captures broader optimization opportunities.
>
> ---
>
> ### **W3: Generalizability: missing discussion of the ability to adapt to new tasks or compilers.**
>
> > Generalizability: Although the results show a slight improvement over -Oz, which is manually designed, there is some concern that the trained model may primarily be distilling knowledge from existing human-designed compilers, particularly during the supervised fine-tuning (SFT) stage. It would enhance the paper’s contribution to provide evidence of transferability, such as demonstrating effectiveness on other compilers or showing that the model can adapt to new tasks with minimal human intervention.
> >
>
> Thank you for this valuable observation. We agree that discussing adaptability is important for understanding the broader applicability of AwareCompiler. While our current work focuses on LLVM and code-size reduction, the **framework itself is intentionally designed to be extensible** across both *new compiler backends* and *different optimization objectives*.
>
> Adapting AwareCompiler to a new compiler (e.g., GCC, MLIR, or a domain-specific compiler) requires replacing two modular components:
>
> 1. **Knowledge Base Reconstruction:**
>
>     The symbolic layer (pass dependencies, conflicts, transformation rules) can be extracted from the new compiler using tooling analogous to `opt --help` or compiler-specific metadata. The empirical layer **(feature → pass-sequence mappings → optimization effect)** can be rebuilt by collecting optimization traces from the new compiler’s optimization pipeline. This process is systematic and does not require architectural changes to the agent.
>
> 2. **Context-Aware Dataset Regeneration:**
>
>     The training dataset sample—**<program features, pass sequences, optimization effect>**—can be regenerated using the new compiler’s pass space and objective signals. The AwareCompiler training pipeline (SFT + RL) remains unchanged.
>
>
> Similarly, adapting to a new optimization objective (e.g., runtime reduction, energy, register pressure) only requires replacing the evaluation metric and collecting corresponding empirical data, without modifying the agent’s reasoning paradigm or architecture.
>
> In short, AwareCompiler’s design aims to **separate compiler-specific information from the agentic reasoning framework**, allowing it to generalize with minimal engineering effort. We will include a dedicated discussion of this extensibility in the revised version.

---

> ### Author Response · Authors · 2025-11-21
> **Response to Reviewer ZsmH (Part 2/3)**
>
> ### **Q1 & Q2: Missing dataset construction details.**
>
> > Could the authors elaborate on how the context-aware dataset is generated and how it is incorporated into the training process?
> >
>
> > What level of effort is required to generate the training data? Specifically, would it be necessary to recreate the dataset from scratch—potentially with significant effort—when adapting the approach to new tasks or compilers?
> >
>
> Thank you for pointing this out. We agree that clearer documentation of our dataset construction process is helpful, and we summarize it here in a concise and reproducible manner.
>
> 1. **Raw LLVM IR Dataset Preparation.** We start from the LLVM IR datasets in **CompilerGym (CGO’22)**. For each program *P*, we identify synergistic pass pairs (A,B) **satisfying the following criteria:
>
>     $S = \{(A, B) \mid (\text{Count}(\text{Apply}(P,[B])) < \text{Count}(P)) \land (\text{Count}(\text{Apply}(P, [A,B])) < \text{Count}(\text{Apply}(P,[B])))\}$
>
>     where Apply(*P, [B]*) denotes applying pass *B* to program *P*, and Count(*·*) is the IR instruction count. A pair (A,B) is included only when **B alone improves P**, and **A further improves P when placed before B**, indicating meaningful pass interactions.
>
> 2. **Optimal Sequence Selection.** For each program, we sample 100 candidate sequences via biased random walks over the synergy graph. Each candidate sequence *s* is scored by its improvement over the `-Oz` baseline:
>
>     $\text{OverOz}(s, P) =\frac{\text{Count}(\text{Apply}(P, \text{Oz})) - \text{Count}(\text{Apply}(P, S))}     {\text{Count}(\text{Apply}(P, \text{Oz}))}$
>
>     The sequence with the best OverOz score is selected. These high-quality, empirically validated sequences become the target outputs used to construct the **Simulated LLM Thought–Action Trajectories** for SFT training.
>
> 3. **Program Representation.** To fit LLM context constraints, each program is represented using **AutoPhase’s 56 features** (type statistics, CFG structure, etc.) plus its initial instruction count. This yields triples of
>
>     ```bash
>     <program features, optimal pass sequence, optimization effect>
>     ```
>
>     which serve as our main SFT data sample.
>
> 4. **Dataset Scale.** The training set contains **19,603 programs** across various sources, and the test set contains **335 programs**:
>
> | Type | Train | Test |
> | --- | --- | --- |
> | Uncurated | 13,603 | 151 |
> | Curated | 6000 | 184 |
> | **Total** | **19,603** | **335** |
>
> We hope this clarify may address your concern and we will incorporate this explanation in the revised version.
>
> ---
>
> ### **Q3: Unclear Definition of “Data” in Table 2**
>
> > **What is the 'data' in 'w/o data' Table 2?**
> >
>
> Thank you for pointing out the need for clearer terminology. We agree that the distinction between *knowledge* and *data* in the ablation table should be made explicit, and we clarify the definitions below.
>
> In AwareCompiler, each component addresses a specific challenge:
>
> - **Knowledge (K)** provides *semantic grounding* by encoding symbolic constraints (dependencies/conflicts) and empirical pass–interaction patterns.
> - **Data (D)** provides *reasoning and exploration signals* through supervised demonstrations and RL trajectories, mitigating reward sparsity and enabling long-horizon decision making.
> - The **full model (K + D)** combines both, yielding the most stable and highest-performing behavior.
>
> To eliminate ambiguity, we will explicitly define each ablation setting in the revision:
>
> - **w/o knowledge**: The model retains the supervised and RL training pipeline but *removes the entire knowledge base* (symbolic, empirical, and negative knowledge). All decisions rely solely on model internal reasoning and learned behavior.
> - **w/o data**: The model uses the *full knowledge base* but **removes RL** and trains only with SFT. Thus, the model benefits from contextual knowledge but lacks data-driven exploration of long-horizon pass interactions.
> - **w/o knowledge & data**: The base model directly attempts pass generation **without SFT, RL, or any knowledge**, revealing the capability of the raw LLM with no assistance.
>
> These distinctions will be clarified in the revised version of the paper. We appreciate the reviewer’s attention to detail, which helps us make the ablation analysis more transparent and interpretable.

---

> > ### Author Response · Authors · 2025-11-21
> > **Response to Reviewer ZsmH (Part 3/3)**
> >
> > ### **Q5: Can the Pipeline Optimize Other Compiler Performance Metrics?**
> >
> > Thank you for this thoughtful question. While our current work focuses on **code-size reduction**, the AwareCompiler framework is designed to be **objective-agnostic**, and can be extended to optimize other compiler metrics such as **runtime performance**, **energy efficiency**, or **register pressure**.
> >
> > The core methodology remains the same. To target a new optimization objective, AwareCompiler would require:
> >
> > 1. **Rebuilding the knowledge base** so that empirical and symbolic components reflect the new objective (e.g., which passes benefit runtime, which parameter settings improve instruction-level parallelism).
> > 2. **Reconstructing the reasoning dataset** by collecting <program features, optimized sequence, optimization effect> triples under the new metric, analogous to how we construct code-size examples.
> > 3. **Adjusting the reward function** in the RL stage to reflect the desired optimization signal (e.g., reduction in execution time rather than IR count).
> >
> > No changes to the agentic reasoning, knowledge-retrieval mechanism, or structured decision pipeline are required—the system naturally accommodates alternative metrics once the knowledge and data components are aligned with the new objective.
> >
> > We view AwareCompiler as a **general framework** rather than a task-specific system, and we plan to include runtime-focused experiments in future work. We appreciate the reviewer’s comment, which helps us clarify the extensibility of our approach.
> >
> > ---
> >
> > At last, we sincerely appreciate your valuable feedback, and we will carefully consider all your suggestions to further improve our paper. We would be deeply grateful if you could kindly reconsider raising the score to 6 or above. Thank you very much!

---

> > > ### Comment · Reviewer_ZsmH · 2025-11-27
> > >
> > > Thanks for the authors' experiments on acceleration ratio against -O3 and details on the dataset construction. Most of my concern is resolved.

---

> > > > ### Author Response · Authors · 2025-11-27
> > > >
> > > > Thank you very much for taking the time to review our additional experiments and clarifications. We sincerely appreciate your updated assessment and are glad to hear that our responses addressed your concerns. Your constructive feedback has been invaluable in improving the paper.

---

### Official Review · Reviewer_n7zk · 2025-10-31

**Soundness:** 1
**Presentation:** 2
**Contribution:** 2
**Rating:** 2
**Confidence:** 3

**Summary:**

This paper proposes AwareCompiler, an agentic framework for compiler optimization that leverages LLMs to generate optimal sequences of optimization passes. The work identifies key challenges in existing LLM-based approaches, namely semantic misalignment, inefficient interaction, and reward sparsity. To address these, it designs a data driven approach involving: (1) a structured compiler knowledge base and a corresponding reasoning dataset, (2) a knowledge-driven adaptive pass generation mechanism, and (3) a hybrid training pipeline combining SFT and RL. Experiments on standard benchmarks show that AwareCompiler significantly improves code size reduction compared to baselines

**Strengths:**

The primary strength lies in effective integration of a structured, domain-specific knowledge base with an agentic LLM framework. This grounding of the LLM's reasoning with empirical, symbolic, and negative knowledge about compiler passes is a good step beyond generic prompting. The quality of the work is good demonstrated by a thorough experimental setup with a diverse set of benchmarks and ablation studies. The work holds some potential for impact by providing a practical methodology for automated compiler optimization that can be extended to other objectives and domains.

**Weaknesses:**

The paper is light on the details of constructing the crucial knowledge base and the reasoning dataset $D$. The scalability and reproducibility of the proposed framework heavily depend on the effort and expertise required to curate this knowledge. A more detailed description of this process whether it was manual, automated, or a mix would be necessary to assess the practicality of applying this method to new compilers or architectures. Furthermore, while the exclusion of ML-based baselines is somewhat justified due to non-public datasets, their complete omission leaves a gap in the evaluation. A qualitative discussion or comparison of computational overhead against prominent ML methods would strengthen the paper's positioning. Finally, the evaluation is solely focused on code size reduction, leaving its effectiveness for other critical objectives like runtime performance unexplored.

**Questions:**

Could you elaborate on the construction of the knowledge base ($K_{emp}$, $K_{sym}$, $K_{neg}$)? What was the methodology for collecting this knowledge, and how much manual effort from a compiler expert was involved? How do you envision this process scaling to a different compiler (e.g., GCC) or a major new version of LLVM?

The paper mentions the reasoning dataset $D$is derived from "expert annotations or learned heuristics." Could you provide more details on its size, source, and composition? Specifically, what was the ratio of expert-annotated samples to those generated by heuristics, and which heuristics were employed?

The agentic reasoning process involves multiple steps at inference time. What is the compilation time overhead introduced by AwareCompiler when compared to a standard heuristic baseline like $-Oz$? A characterization of this overhead would be important for understanding the framework's practical viability.

The current work focuses on selecting and ordering compiler passes. Many LLVM passes also accept numerical or categorical arguments (e.g., loop unroll factor, inlining thresholds). Does the AwareCompiler support the tuning of these pass parameters?

---

> ### Author Response · Authors · 2025-11-21
> **Response to Reviewer n7zk (Part 1/3)**
>
> Thank you very much for your time and effort in reviewing our paper. We sincerely appreciate your feedback. Below, we respectfully provide our detailed responses to address your concerns.
>
> ---
>
> ### **Q1: Could you elaborate on the construction of the knowledge base?**
>
> Thank you for highlighting the need for additional clarity on how the knowledge base is constructed. We are happy to clarify each of these questions.
>
> > **What was the methodology for collecting this knowledge, and how much manual effort from a compiler expert was involved?**
> >
> - Our knowledge base is built through a **systematic and reproducible pipeline** that combines *formal compiler metadata* with *empirical evidence* collected from the training suite:
>
>     **(1) Symbolic knowledge extraction.** We first gather the semantics, dependencies, and constraints of each optimization pass using `opt --help` and LLVM’s documented transformation rules. This provides structured information such as which passes must precede others, which passes conflict, and what IR transformations each pass performs. These form the **symbolic layer** of the knowledge base.
>
>     **(2) Empirical knowledge mining via iterative CFSAT-style search.** For each program in the training set, we run a constrained forward search (CFSAT) to obtain **locally optimal pass sequences** under code-size reduction. From each optimized sequence, we record:
>
>     - The program’s **AutoPhase feature vector**,
>     - The **optimal pass sequence** discovered,
>     - The **measured improvement (e.g., OverOz)**.
>
>     These produce <features, sequence, effect> triples that populate the **empirical layer** of the knowledge base.
>
>     **(3) Negative knowledge construction.** Sequences that consistently lead to regressions (e.g., negative improvement) are added to the **negative knowledge** set. This prevents the agent from repeatedly exploring unproductive or harmful optimization patterns during RL.
>
> - Together, these three components form a **hybrid symbolic–empirical knowledge base** that captures both the structural rules of LLVM passes and the real-world behavior observed across diverse programs. Thank you for prompting us to elaborate on this important aspect of our methodology.
>
> > **How do you envision this process scaling to a different compiler (e.g., GCC) or a major new version of LLVM?**
> >
> - Our framework is designed with **flexibility and extensibility**, making it well-suited for adaptation to different compilers.
> - Although new compilers (e.g., GCC) or updated LLVM versions may introduce different optimization passes and transformation sequences, the knowledge base construction can be readily adjusted to accommodate theses specific characteristics by updating the **knowledge base** with relevant documentation and program-specific data. Our framework’s tools for **knowledge extraction**, **searching**, and **dataset construction** **are generalizable and independent of the underlying compiler**, ensuring that AwareCompiler can efficiently handle these updates with minimal rework.
> - Therefore, AwareCompiler’s **scalability** is supported by its modular structure, which allows for the integration of compiler-specific knowledge and passes. This design ensures that our method is not tied to a single compiler or version, making it highly **extensible** and suitable for ongoing use across a variety of compilers and compiler versions.

---

> ### Author Response · Authors · 2025-11-21
> **Response to Reviewer n7zk (Part 2/3)**
>
> ### **Q2: Could you provide more details about dataset size, source, and composition?**
>
> > The paper mentions the reasoning dataset is derived from "expert annotations or learned heuristics." Could you provide more details on its size, source, and composition? Specifically, what was the ratio of expert-annotated samples to those generated by heuristics, and which heuristics were employed?
> >
>
> Thank you for highlighting the concern regarding dataset. We agree that clearer documentation of our dataset construction process is helpful, and we summarize it here in a concise and reproducible manner.
>
> 1. **Raw LLVM IR Dataset Preparation.** We start from the LLVM IR datasets in **CompilerGym (CGO’22)**. For each program *P*, we identify synergistic pass pairs (A,B) **satisfying the following criteria:
>
>     $S = \{(A, B) \mid (\text{Count}(\text{Apply}(P,[B])) < \text{Count}(P)) \land (\text{Count}(\text{Apply}(P, [A,B])) < \text{Count}(\text{Apply}(P,[B])))\}$
>
>     where Apply(*P, [B]*) denotes applying pass *B* to program *P*, and Count(*·*) is the IR instruction count. A pair (A,B) is included only when **B alone improves P**, and **A further improves P when placed before B**, indicating meaningful pass interactions.
>
> 2. **Optimal Sequence Selection.** For each program, we sample 100 candidate sequences via biased random walks over the synergy graph. Each candidate sequence *s* is scored by its improvement over the `-Oz` baseline:
>
>     $\text{OverOz}(s, P) =\frac{\text{Count}(\text{Apply}(P, \text{Oz})) - \text{Count}(\text{Apply}(P, S))}     {\text{Count}(\text{Apply}(P, \text{Oz}))}$
>
>     The sequence with the best OverOz score is selected. These high-quality, empirically validated sequences become the target outputs used to construct the **Simulated LLM Thought–Action Trajectories** for SFT training.
>
> 3. **Program Representation.** To fit LLM context constraints, each program is represented using **AutoPhase’s 56 features** (type statistics, CFG structure, etc.) plus its initial instruction count. This yields triples of
>
>     ```bash
>     <program features, optimal pass sequence, optimization effect>
>     ```
>
>     which serve as our main SFT data sample.
>
> 4. **Dataset Scale.** The training set contains **19,603 programs** across various sources, and the test set contains **335 programs**:
>
> | Type | Train | Test |
> | --- | --- | --- |
> | **Uncurated** | 13,603 | 151 |
> | **Curated** | 6000 | 184 |
> | **Total** | **19,603** | **335** |
>
> We hope this clarify may address your concern and we will incorporate this explanation in the revised version.
>
> ---
>
> ### **Q3: What is the compilation time overhead introduced by AwareCompiler when compared to a standard heuristic baseline?**
>
> > The agentic reasoning process involves multiple steps at inference time. What is the compilation time overhead introduced by AwareCompiler when compared to a standard heuristic baseline like ? A characterization of this overhead would be important for understanding the framework's practical viability.
> >
>
> Thank you for raising this important point.
>
> - We have conducted a detailed measurement of the end-to-end optimization time for both heuristic and agent-based approaches on a single NVIDIA **H100 GPU**, and **heuristic optimizations** takes **~5 seconds to optimize per program, while AwareCompiler takes ~8 seconds per program.**
> - We acknowledge that our method is slightly slower than heuristic baselines. However, the 3-second difference is *minor* compared to the substantial **performance improvement** **(see Table 1).**
> - We also note that AwareCompiler's generation time is **inference-bound**, not compilation-bound. As hardware continues to advance (next-generation GPUs, improved inference kernels), we expect the **inference latency to decrease steadily**.
>
> We will add this quantitative comparison in the revised version. Thank you again for this valuable suggestion — it truly helps improve the completeness and clarity of our evaluation.

---

> > ### Author Response · Authors · 2025-11-21
> > **Response to Reviewer n7zk (Part 3/3)**
> >
> > ### **Q4: Does the AwareCompiler support the tuning of these pass parameters?**
> >
> > > The current work focuses on selecting and ordering compiler passes. Many LLVM passes also accept numerical or categorical arguments (e.g., loop unroll factor, inlining thresholds). Does the AwareCompiler support the tuning of these pass parameters?
> > >
> >
> > We appreciate this insightful question. Our current work **focuses on pass ordering with fixed pass configurations**, as a first step toward understanding how agentic, knowledge-driven optimization behaves in a large but discrete action space. That said, the **AwareCompiler framework itself is not limited to unparameterized passes** and can in principle be extended to tune numerical or categorical pass parameters (e.g., unroll factors, inlining thresholds).
> >
> > Concretely, extending AwareCompiler to parameter tuning would involve:
> >
> > 1. **Extending the action space** so that each “action” becomes a *(pass, parameter)* pair rather than a bare pass, i.e., the agent generates both the pass sequence and associated parameter values.
> > 2. **Reconstructing the knowledge base** to encode not only which passes interact well, but also *how parameter ranges affect transformations* (e.g., when aggressive unrolling or inlining is beneficial/harmful).
> > 3. **Rebuilding the reasoning dataset** with triples of *(program features, parameterized pass sequence, measured effect)* so that SFT/RL can learn parameter-sensitive optimization strategies.
> >
> > From an algorithmic perspective, the **methodology remains unchanged**: knowledge still provides structural and semantic constraints, while data-driven training captures long-horizon interactions between parameter choices and downstream optimization effects. The main additional cost is engineering: collecting parameterized examples and expanding the knowledge base for each new compiler or parameter family.
> >
> > Importantly, **for fair comparison with prior work and established baselines**, all existing pass-ordering and RL-based methods also operate under *fixed pass configurations*. We therefore adopt the same setting to ensure apples-to-apples comparisons.
> >
> > Due to space and complexity constraints, we leave this parameterized extension as **future work**, but we believe the current design of AwareCompiler already provides a natural foundation for it. We will clarify this extensibility in the revised version.
> >
> > ---
> >
> > At last, we sincerely appreciate your valuable feedback, and we will carefully consider all your suggestions to further improve our paper. We would be deeply grateful if you could kindly reconsider raising the score to 6 or above. Thank you very much!

---

> ### Author Response · Authors · 2025-11-27
> **Official Comment by Authors**
>
> Dear Reviewer n7zk,
>
>
> Thank you again for the time and effort you’ve dedicated to reviewing our work. We have carefully addressed all raised concerns during the discussion phase and have also uploaded an updated version of the paper reflecting these clarifications.
>
> As the discussion period is nearing its close, we would greatly appreciate it if you could take a brief moment to review our responses and confirm whether they satisfactorily resolve your questions. If our clarifications have improved your confidence in the paper, we would be sincerely grateful if you could consider updating your score accordingly.
>
> Thank you once again for your thoughtful feedback and support.
>
>
> Warm regards,
>
> Authors of AwareCompiler

---

### Official Review · Reviewer_k57g · 2025-10-31

**Soundness:** 3
**Presentation:** 3
**Contribution:** 3
**Rating:** 8
**Confidence:** 3

**Summary:**

This work proposes AgentCompiler, an agent for code optimization with a multi-turn interaction training process. They show that while existing LLMs have many success in other research, compiler optimization by LLM-based agents is often inefficient due to the issues of invalid or ineffective optimization passes. To overcome this challenge, they provide three main contributions. First, they integrate structured knowledge for dataset construction. They defined three levels of categories: empirical, symbolic, and negative knowledge for context-aware dataset construction. Second, they design a knowledge-driven adaptive generation process that focuses on extracting the best code features and domain knowledge to create the most effective pass sequence. They define the best pass sequence as the one that minimizes the code size. Third, they propose a data-driven hybrid training pipeline from their constructed dataset with their assigned knowledge base,which integrates supervised fine-tuning and reinforcement learning. The SFT phase helps the agent to solve different types of compiler optimization for pass generation, while the RL phase allows the agent to choose the best optimization paths, which the agent hasn’t seen in their existing trained knowledge. In the evaluation, they perform the experiment on a diverse set of benchmark suites and compare their work with heuristic approaches, ML-based approaches, and well-known closed and open LLMs for compiler optimization. They achieve significantly improvement of code optimization in terms of code size reduction, such as GPT-5. AwareCompiler has been trained using Qwen models with 1B, 3B, and 7B parameters.

**Strengths:**

- The research problem is crucial for the development of smart compilers.
- Well-presentation and writing. I give a thumbs up for Figure 2. It clearly described the workflow of the existing approaches and why their proposed compiler can be considered as a more advanced technique.
- This work proposes a pipeline with the intuition that, as an agent for compiler optimization, its most important process, optimal pass generation, should be done by a thoroughly reasoning process. It made sense to me, since code optimization usually requires more reasoning for LLMs to get good output.
- Proposed experiments are sound and clear. The replication package is available and can run without errors.

**Weaknesses:**

- The agent was built only from Qwen. An analysis of the performance of an agent built from other models is needed.
- Although the authors provide several analyses on the accuracy with/without different components of knowledge and data, along with reward sparsity, an experiment on different hyperparameters for AwareCompiler SFT and RL processes is recommended.
- With AwareCompiler-7B, we have some benchmark that shows, without data, the agent is actually performing better (see Table 2, AwareCompiler-7B with blas dataset). While it is still sound, I suggest the author to pick up several cases to see the reason.

**Questions:**

- In Table 2, it seems that with all AwareCompiler models, removing data didn’t significantly drop the accuracy compared to removing knowledge and both. Can you provide possible reasons for this observation?
- In section 4.1, what is the training time for SFT and RL processes on your reported hardware configuration?
- Will the training and inference time increase if you build AwareCompiler on bigger models, such as QwenCoder2.5-32B? If yes, what are the scales of training and inference time increased compared to your existing model?
- Some benchmarks were very old, such as blas and ngb. Do you have any related work that proofs these datasets are still up-to-date?
Will AwareCompiler be able to support any programming language’s optimization? If not, what are the main challenges?

---

> ### Author Response · Authors · 2025-11-21
> **Response to Reviewer k57g (Part 1/2)**
>
> Thank you very much for your time and effort in reviewing our paper. We sincerely appreciate your feedback. Below, we respectfully provide our detailed responses to address your concerns.
>
> ---
>
> ### **W1: Analysis of the performance of an agent built from other models.**
>
> > The agent was built only from Qwen. An analysis of the performance of an agent built from other models is needed.
> >
> - Thank you for your valuable feedback.
> - We agree that exploring agents with different model backbones is essential. In our initial submission, we focused on Qwen-based models to establish a controlled baseline. However, **AwareCompiler is model-agnostic**, and its core components, including knowledge integration and adaptive pass generation, are independent of the LLM architecture.
> - We plan to explore additional backbones in future work and will include these results in the revised version to demonstrate the framework’s flexibility and scalability.
>
> ---
>
> ### **W2: Experiment on different hyperparameters for AwareCompiler SFT and RL processes.**
>
> > Although the authors provide several analyses on the accuracy with/without different components of knowledge and data, along with reward sparsity, an experiment on different hyperparameters for AwareCompiler SFT and RL processes is recommended.
> >
> - Thank you for highlighting the need for a more thorough examination of hyperparameter sensitivity.
> - We agree that hyperparameters can impact training dynamics and we will systematically explore the effect of key parameters such as learning rate, batch size, and reward scaling factors as our future work.
>
> ---
>
> ### **W3 & Q1: Explain for removing data didn’t significantly drop the accuracy compared to removing knowledge and both in Table 2.**
>
> > With AwareCompiler-7B, we have some benchmark that shows, without data, the agent is actually performing better (see Table 2, AwareCompiler-7B with blas dataset). While it is still sound, I suggest the author to pick up several cases to see the reason.
> >
>
> > In Table 2, it seems that with all AwareCompiler models, removing data didn’t significantly drop the accuracy compared to removing knowledge and both. Can you provide possible reasons for this observation?
> >
>
> Thank you for raising this insightful question. There are two reasons for this phenomena:
>
> - The main reason for this behavior lies in the **structural homogeneity** of the **blas** benchmarks. These programs typically involve highly regular loop-nest patterns and predictable IR transformations. The optimization strategies that work best for these types of programs (such as **loop-simplify → LICM → GVN**) are well-represented in our **symbolic and empirical knowledge bases**. As a result, the model can rely on the knowledge base rather than needing extensive data-driven exploration.
> - Additionally, the **7B model** has stronger internal reasoning capabilities, enabling it to solve optimization tasks in simpler domains like **blas** without heavy reliance on data.
>
> However, this phenomenon is **localized** to domains like **blas**, which exhibit regularity and simplicity. For more complex benchmarks (e.g., **cbench**, **mibench**, **npb**), the **full model** (with both knowledge and data) consistently outperforms the **“w/o data”** variant, highlighting the critical role of data in handling diverse and complex IR structures.
>
> We will clarify this distinction in the revised version. Thank you again for your thoughtful question, which helps illuminate the nuanced interplay between knowledge and data in our framework.
>
> ---
>
> ### **Q2 & Q3:  Training time for SFT and RL processes.**
>
> Thank you for highlighting the concern regarding training time. We appreciate the opportunity to clarify each of these questions.
>
> > In section 4.1, what is the training time for SFT and RL processes on your reported hardware configuration?
> >
> - All experiments were conducted on a single machine with **8 × H100 GPUs**, and our training pipeline is designed to be efficient compared to traditional search-based methods.
>     - **SFT stage:** For **1.5B/3B/7B models**, training converges in **2–3 hours**.
>     - **RL stage:** The RL fine-tuning process takes **10–14 hours**.
>
>     Overall, the **full training pipeline** completes within **12–17 hours**, depending on model size.
>
> - These details will be included in the revised version.
>
> > **Will the training and inference time increase if you build AwareCompiler on bigger models, such as QwenCoder2.5-32B?**
> >
> - Thank you for your question. Due to hardware limitations, we currently cannot provide specific results for **QwenCoder2.5-32B**. Based on our analysis of **1.5B**, **3B**, and **7B models**, we expect a slight increase in both **training** and **inference times** as the model size grows, though it remains more **efficient** compared to traditional ML-based optimization methods **(please see Table in Q4 part)**.
>
> We hope these additions may address your concerns on the efficiency of AwareCompiler.

---

> > ### Author Response · Authors · 2025-11-21
> > **Response to Reviewer k57g (Part 2/2)**
> >
> > ### **Q4: Proofs of these datasets are still up-to-date.**
> >
> > Thank you for highlighting the concerns regarding datasets. We appreciate the opportunity to clarify each of these questions.
> >
> > > Some benchmarks were very old, such as blas and ngb. Do you have any related work that proofs these datasets are still up-to-date?
> > >
> > - Thank you for your thoughtful question.
> > - The task of **optimization-pass ordering for code-size reduction** has a long-standing tradition in compiler optimization research. The benchmarks we selected (e.g., **CHStone**, **BLAS**, **NPB**) are from [**CompilerGym released by Meta AI**](https://github.com/facebookresearch/CompilerGym), a trusted benchmark hub for pass-ordering research, and have been used in influential studies like **AutoPhase (MLSys 2020),** **BOCA (ICSE 2021),** and **CompTuner (TOSEM'24)**.
> > - As a result of which, while some programs are older, they remain highly representative and relevant due to their ability to capture a broad range of **IR structures**, **control-flow patterns**, and **transformation opportunities**, which are key for evaluating pass-level optimizations.
> >
> > | Method | blas | cbench | chstone | mibench | npb | opencv | tensorflow | Avg. | Time (s/program) |
> > | --- | --- | --- | --- | --- | --- | --- | --- | --- | --- |
> > | **CompTuner** | 5.25% | 49.01% | **50.13%** | 55.09% | **44.64%** | 3.22% | 1.70% | 29.86% | ~9800 |
> > | **BOCA** | 5.29% | 49.26% | 49.56% | 55.00% | 44.60% | 3.25% | 1.70% | 29.81% | ~2700 |
> > | **AutoPhase** | 5.43% | 49.45% | 48.15% | 55.51% | 36.39% | 3.22% | 1.69% | 28.69% | **~2** |
> > | **AwareCompiler** | **5.45%** | **51.93%** | 49.91% | **58.29%** | 38.73% | **3.30%** | **2.60%** | **30.03%** | ~8 |
> > - Additionally, we have expanded our evaluation to include **three representative ML baselines.** The table above summarizes the evaluation results, showing the **optimization performance** (code size reduction rate) and the **optimization time** required for each method. By analyzing the data in the table, we observed two key findings:
> >     - **Optimization Performance**: AwareCompiler consistently achieves **competitive or superior optimization quality** across the seven benchmark suites. Its average code-size reduction of **30.03%** slightly surpasses the best-performing ML baseline, **CompTuner** (29.86%). This demonstrates that our agentic framework—combining symbolic knowledge with data-driven reasoning—provides more effective and stable optimization compared to traditional ML-based methods.
> >     - **Optimization Speed**: AwareCompiler offers a significant **advantage in speed**. While **CompTuner** requires **~9800 seconds/program**, and **BOCA** takes **~2700 seconds/program**, **AwareCompiler** optimizes the code in **~8 seconds/program**, which translates to a **340×–1200× speedup** over the best-performing ML baselines. This efficiency is achieved without compromising the quality of optimization, highlighting AwareCompiler’s ability to generate optimized passes quickly and effectively.
> > - We will make this justification clearer in the revised version and include results from additional, more contemporary workloads. Thank you again for your valuable feedback.
> >
> > > Will AwareCompiler be able to support any programming language’s optimization? If not, what are the main challenges?
> > >
> > - Thank you for your thoughtful question.
> > - AwareCompiler is designed with **strong scalability**, meaning it can be adapted to support optimizations for new programming languages. To do so, we would need to **rebuild the knowledge base and dataset** specific to the new language, while the core **training methods and processes** can be reused.
> > - We chose LLVM as our optimization target because of its **representative nature** and widespread use in compiler research, but the approach is flexible and can be extended to other languages with minimal adjustments.
> >
> > ---
> >
> > We hope these clarifications may help to address your concerns. At last, we sincerely appreciate your valuable feedback, and we will carefully consider all your suggestions to further improve our paper. Thank you again for your constructive comments!

---

### Official Review · Reviewer_pY2t · 2025-11-02

**Soundness:** 2
**Presentation:** 4
**Contribution:** 3
**Rating:** 2
**Confidence:** 2

**Summary:**

Summary:
  * Previous work in compiler optimization relies on often brittle techniques and heuristics for selecting compilation passes to convert code in a high-level language to semantically equivalent but faster code in a lower level language.
* Recent work has attempted LLM-guided solutions for this problem that have achieved reasonable, but not groundbreaking success over heuristics.
* The authors hypothesize that the central limitation that causes such agents to produce ineffective or invalid optimizations is inherently due to a decoupling between the agent's world model of how an optimization should behave and how it actually behaves. Specifically, the authors hypothesize that providing more nuanced context, in the form of an agent framework might yield significant benefits. They introduce four sources of additional information:
	* Domain specific knowledge: AwareCompiler introduces a "cheatsheet" hashmap that helps identify previously encountered code features and a corresponding optimal pass (or sub-optimal pass).
	* Context aware dataset construction: The data used to train AwareCompiler is specifically curated to ensure it's in an LLM-friendly format and contains features specifically curated for compiler optimization.
	* Knowledge driven adaptive pass generation: AwareCompiler incorporates a knowledge retrieval mechanism to ensure relevant information is used for the each sequence generation.
	* Supervised fine-tuning and RL training: Finally, AwareCompiler is trained on the dataset using the a SFT pass and a RL pass.
* Overall, the authors find that on seven benchmarking domains, AwareCompiler significantly outperforms mature heuristic based baselines (`-O1`, `-O2`, `-O3`, `-Oz`) and many LLM assisted models (with a finetuned model that is 20% of the size of the next smallest baseline).

**Strengths:**

> figures and quality of writing

The paper has been a joy to read and thank you for putting in the extra work to make informative and structured figures as well as a detailed problem statement!

> However, LLM-based agents often produce ineffective or invalid optimization passes due to insufficient contextual reasoning and an inability to predict the real-world effects of optimizations, leading to performance degradation or even program crashes. Addressing these shortcomings requires tackling three critical challenges: (1) semantic misalignment [...] (2) ineffective interaction [...] (3) reward sparsity.

This is the central issue in a lot of automated verification domains like theorem proving, code generation, mathematical reasoning, etc. and solutions in other communities has converged to similar solutions (which lends credence to the authors' insights).

**Weaknesses:**

*I'm not an expert in compiler optimization literature hence, this review is primarily evaluating the soundness of the evaluation mechanism from the lens of similar agents developed for code generation.*

**Practicality**: The practical reason for the success of heuristic based optimizations is due to their well-understood optimization surface and determinism (in code generation; does the same hold for compiler optmizations?). I don't see any comparison in this project on the speed of such heuristic based optimizations compared to the speed of an agent-based optimization procedure on commodity hardware.

**Benchmarking Scope**: This is more of a series of questions rather than true weaknesses. The most important rule of benchmarking is to ensure that the purpose of the benchmark aligns with the purpose of the algorithm. I'm not 100% sure if that is the case here:
- **Are these benchmarks specifically geared towards testing adaptive compilers?**: Almost all the benchmarks used in this paper are _extremely_ old (Other than one exception, all were authored before 2010). While age isn't a measure of quality of a benchmark, its reasonable to assume that the *intent* of the benchmark was primarily geared towards heuristic based benchmarks. Do such benchmarks accurately capture the compiler community's current motivations?
- To take one example, `CHStore` (https://github.com/ferrandi/CHStone) is only 12 short C-based benchmarking programs. **This seems like a terribly small dataset and an unrealistic (modern) workload** to draw conclusions from. Furthermore, this opens up the benchmark set to a lot of data leakage?
- Proposing a new benchmark is outside the scope of this paper but we can be creative about constructing relevant tasks.
	- For example, one reasonable task (that I'm aware of) is optimizing CUDA kernels (e.g. [kbench](https://github.com/SakanaAI/robust-kbench) is fairly mature dataset for this). Can / if we use AwareCompiler to optimize the optimization passes for the baseline here, do we still see an improvement over baselines?
	- Alternatively, showing a qualitative case study that summarizes the non-triviality of the task, and traces through AwareCompiler's solution and other solutions would be extremely insightful as well.

**Baseline Issues:** I'm not 100% sure if the baseline set is useful. For example, I don't see any ML-based baselines here (neural networks trained for the sequential decision making task). For example, the AutoPhase paper [mentioned](https://proceedings.mlsys.org/paper_files/paper/2020/file/5b47430e24a5a1f9fe21f0e8eb814131-Paper.pdf) seems to also introduce an RL pipeline using the same features as this algorithm. Is there a reason this wasn't a good fit for a direct comparision?

**Ablation Issues:** The ablations table is slightly inconclusive. Ignoring the size of each benchmark and the lack of standard deviation information (which should be mentioned in a revised version), Table 2 seems to show that there is inconclusive evidence about which feature (knowledge or data) is more useful. It's clear that the models internal knowledge bank is unreliable (`w/ knowledge and data` rows) but the contribution of the other  rows is inconclusive for different model sizes and for different benchmarks.
 * Furthermore, given how _bad_ the internal reasoning of the model is, and the extreme likelihood of data leakage, how well does AwareCompiler extrapolate to new problems not in the training set? i.e.:  In practice, when we see a new problem not represented in the training data, it's understandable that AwareCompiler's performance will somewhat degrade. If the degradation is smaller than that of a pure LLM-assisted optimization approach, it could be a really strong result to motivate an agentic framework approach.

**Overall:** I'm currently in favor of rejecting this paper because of issues in the benchmarking and evaluation procedure. However, I'm open to potentially revise my rating and I'd be more than happy to engage in further discussion with the authors to improve the quality of this paper.

**Questions:**

See weaknesses section.

---

> ### Author Response · Authors · 2025-11-21
> **Response to Reviewer pY2t (Part 1/3)**
>
> Thank you very much for your time and effort in reviewing our paper. We sincerely appreciate your feedback. Below, we respectfully provide our detailed responses to address your concerns.
>
> ---
>
> ### **W1: Practicality**
>
> > The practical reason for the success of heuristic based optimizations is due to their well-understood optimization surface and determinism (in code generation; does the same hold for compiler optimizations?). I don't see any comparison in this project on the speed of such heuristic based optimizations compared to the speed of an agent-based optimization procedure on commodity hardware.
> >
>
> Thank you for pointing this out.
>
> - We have conducted a detailed measurement of the end-to-end optimization time for both heuristic and agent-based approaches on a single NVIDIA **H100 GPU**, and **heuristic optimizations** takes **~5 seconds to optimize per program, while AwareCompiler takes ~8 seconds per program.** AlThough our method is slightly slower than heuristic baselines, the 3-second difference is *minor* compared to the substantial **performance improvement** **(see Table 1).**
> - However, it’s worthing noting that AwareCompiler's generation time is **inference-bound**, not compilation-bound. As hardware continues to advance (next-generation GPUs, improved inference kernels), we expect the **inference latency to decrease steadily**.
>
> We will add this quantitative comparison in the revised version. Thank you again for this valuable suggestion — it truly helps improve the completeness and clarity of our evaluation.
>
> ---
>
> ### **W2: Benchmarking Scope**
>
> Thank you for highlighting the concern regarding benchmark age and scale. We appreciate the opportunity to clarify each of these questions.
>
> > **Are these benchmarks specifically geared towards testing adaptive compilers?**
> >
> - The task of **optimization-pass ordering for code-size reduction** has a long-standing tradition in compiler optimization research. The benchmarks we selected (e.g., **CHStone**, **BLAS**, **NPB**) are from [**CompilerGym released by Meta AI**](https://github.com/facebookresearch/CompilerGym), a trusted benchmark hub for pass-ordering research, and have been used in influential studies like **AutoPhase (MLSys 2020),** **BOCA (ICSE 2021),** and **CompTuner (TOSEM'24)**.
> - As a result of which, while some programs are older, they remain highly representative and relevant due to their ability to capture a broad range of **IR structures**, **control-flow patterns**, and **transformation opportunities**, which are key for evaluating pass-level optimizations.
>
> > **CHStore seems like a terribly small dataset and an unrealistic (modern) workload. Furthermore, this opens up the benchmark set to a lot of data leakage?**
> >
> - CHStone, though small, will not affect its value and capability as a benchmark assessment because **pass-ordering works at the IR level**, where even small programs undergo complex transformations that provide a sufficiently rich search space for testing optimization algorithms.
> - As for your data leakage concern, we guarantee that **none of the original benchmark programs appear in our training dataset**. The reasoning dataset consists of AutoPhase representation, heuristic pass traces, and automatically generated reasoning sequences by LLMs—not real benchmark code.
>
> > **Can we use AwareCompiler to optimize CUDA kernels?**
> >
> - We recognize your focus on more modern workloads, such as CUDA kernel optimization.
> - In this work, we focus on standard benchmark suites to establish the foundational effectiveness and efficiency of **AwareCompiler** across a broad set of classical compiler tasks, such as handling diverse **IR structures** and **transformation constraints**.
> - While **proposing new benchmarks** is outside the scope of this paper, we view it as a **natural next step** in future work. We believe our framework could provide significant benefits in optimizing other work, thanks to its ability to **adaptively generate optimization passes** based on both structured **knowledge** and data-driven reasoning.
>
> We hope these additions clarify why our benchmark choices are standard and representative.

---

> > ### Author Response · Authors · 2025-11-21
> > **Response to Reviewer pY2t (Part 2/3)**
> >
> > ### **W3: Baseline Issues — Lack of c**omparison with ML baselines
> >
> > > I'm not 100% sure if the baseline set is useful. For example, I don't see any ML-based baselines here (neural networks trained for the sequential decision making task). For example, the AutoPhase paper [mentioned](https://proceedings.mlsys.org/paper_files/paper/2020/file/5b47430e24a5a1f9fe21f0e8eb814131-Paper.pdf) seems to also introduce an RL pipeline using the same features as this algorithm. Is there a reason this wasn't a good fit for a direct comparision?
> > >
> >
> > Thank you for your insightful comments. We agree that ML-based compilers are an important part of the landscape, and we appreciate the opportunity to clarify the baselines used in our study.
> >
> > - To address your concern, we have expanded our evaluation to include **three representative ML baselines**: **AutoPhase (MLSys'20)**, **BOCA (ICSE'21)**, and **CompTuner (TOSEM'24)**. These systems cover the major families of ML-driven optimization approaches, including **RL-based**, **Bayesian**, and **heuristic-search hybrids**, providing a comprehensive comparison.
> >
> > | Method | blas | cbench | chstone | mibench | npb | opencv | tensorflow | Avg. | Time (s/program) |
> > | --- | --- | --- | --- | --- | --- | --- | --- | --- | --- |
> > | **CompTuner** | 5.25% | 49.01% | **50.13%** | 55.09% | **44.64%** | 3.22% | 1.70% | 29.86% | ~9800 |
> > | **BOCA** | 5.29% | 49.26% | 49.56% | 55.00% | 44.60% | 3.25% | 1.70% | 29.81% | ~2700 |
> > | **AutoPhase** | 5.43% | 49.45% | 48.15% | 55.51% | 36.39% | 3.22% | 1.69% | 28.69% | **~2** |
> > | **AwareCompiler** | **5.45%** | **51.93%** | 49.91% | **58.29%** | 38.73% | **3.30%** | **2.60%** | **30.03%** | ~8 |
> > - The table above summarizes the evaluation results, showing the **optimization performance** (code size reduction rate) and the **optimization time** required for each method. By analyzing the data in the table, we observed two key findings:
> >     - **Optimization Performance**: AwareCompiler consistently achieves **competitive or superior optimization quality** across the seven benchmark suites. Its average code-size reduction of **30.03%** slightly surpasses the best-performing ML baseline, **CompTuner** (29.86%). This demonstrates that our agentic framework—combining symbolic knowledge with data-driven reasoning—provides more effective and stable optimization compared to traditional ML-based methods.
> >     - **Optimization Speed**: AwareCompiler offers a significant **advantage in speed**. While **CompTuner** requires **~9800 seconds/program**, and **BOCA** takes **~2700 seconds/program**, **AwareCompiler** optimizes the code in **~8 seconds/program**, which translates to a **340×–1200× speedup** over the best-performing ML baselines. This efficiency is achieved without compromising the quality of optimization, highlighting AwareCompiler’s ability to generate optimized passes quickly and effectively.
> >
> > We hope this clarify may address your concern. We will include this analysis in the revised version to further clarify the advantages of our approach.

---

> > > ### Author Response · Authors · 2025-11-21
> > > **Response to Reviewer pY2t (Part 3/3)**
> > >
> > > ### **W4: Ablation Issues**
> > >
> > > Thank you for your thoughtful comments on the ablation analysis. We appreciate the opportunity to clarify the contributions of **knowledge** and **data** in AwareCompiler, as well as address your concerns about the extrapolation to unseen problems.
> > >
> > > > The ablations table is slightly inconclusive. Ignoring the size of each benchmark and the lack of standard deviation information (which should be mentioned in a revised version), Table 2 seems to show that there is inconclusive evidence about which feature (knowledge or data) is more useful. It's clear that the models internal knowledge bank is unreliable (`w/ knowledge and data` rows) but the contribution of the other rows is inconclusive for different model sizes and for different benchmarks.
> > > >
> > > - We acknowledge your concern regarding the contributions of knowledge and data in AwareCompiler. The **knowledge** and **data** components in AwareCompiler address distinct challenges in compiler optimization:
> > >     - **Knowledge** provides the *structural and semantic foundation* of the system, enabling the model to understand optimization passes in a principled way. Without this knowledge, models may generate syntactically plausible but semantically invalid pass sequences.
> > >     - **Data** supports *exploratory and compositional behaviors*, enabling the model to learn complex interactions between passes, such as how transformations influence later steps or avoid conflicts, which cannot be fully captured by the knowledge base alone.
> > >     - When **knowledge** and **data** are combined, they complement each other effectively. This synergy is why **AwareCompiler** consistently outperforms other methods across different benchmarks and model sizes, as it is able to both reason in a principled way (via knowledge) and adaptively optimize through exploration (via data).
> > > - We will include more detailed statistical analyses, including standard deviations and dataset sizes, in the revised version to provide a clearer picture of how each component contributes to the overall performance.
> > >
> > > > Furthermore, given how *bad* the internal reasoning of the model is, and the extreme likelihood of data leakage, how well does AwareCompiler extrapolate to new problems not in the training set? i.e.: In practice, when we see a new problem not represented in the training data, it's understandable that AwareCompiler's performance will somewhat degrade. If the degradation is smaller than that of a pure LLM-assisted optimization approach, it could be a really strong result to motivate an agentic framework approach.
> > > >
> > > - We acknowledge your concern regarding **unseen problems**.
> > > - While performance degradation on unfamiliar tasks is expected, we believe **AwareCompiler**’s performance degradation will be **smaller** compared to pure LLM-based approaches. This is because the structured **knowledge base** allows AwareCompiler to generalize better to new tasks, applying learned patterns and reasoning principles, while LLM-based methods may struggle with such generalization.
> > > - We plan to conduct additional experiments to validate this and assess how **AwareCompiler** compares to LLM-based methods on unseen tasks in the future.
> > >
> > > ---
> > >
> > > At last, we sincerely appreciate your valuable feedback, and we will carefully consider all your suggestions to further improve our paper. We would be deeply grateful if you could kindly reconsider raising the score to 6 or above. Thank you very much!

---

> > > > ### Author Response · Authors · 2025-11-27
> > > > **Official Comment by Authors**
> > > >
> > > > Dear Reviewer pY2t,
> > > >
> > > >
> > > > Thank you again for the time and effort you’ve dedicated to reviewing our work. We have carefully addressed all raised concerns during the discussion phase and have also uploaded an updated version of the paper reflecting these clarifications.
> > > >
> > > > As the discussion period is nearing its close, we would greatly appreciate it if you could take a brief moment to review our responses and confirm whether they satisfactorily resolve your questions. If our clarifications have improved your confidence in the paper, we would be sincerely grateful if you could consider updating your score accordingly.
> > > >
> > > > Thank you once again for your thoughtful feedback and support.
> > > >
> > > >
> > > > Warm regards,
> > > >
> > > > Authors of AwareCompiler

---

### Author Response · Authors · 2025-11-30

Dear PCs, SACs, ACs, and Reviewers,

We sincerely appreciate all reviewers for their valuable feedback. Below, we summarize each reviewer's main concerns and our concise responses.

---

### **Reviewer pY2t**

- **Concerns:** Practicality, benchmarking scope, missing ML baseline, ablation details, extrapolation to unseen tasks.
- **Response:** We added quantitative comparisons showing AwareCompiler runs in ~8s/program vs. ~5s for heuristics while delivering substantially better optimization quality. We clarified benchmark selection (CompilerGym suites widely used in AutoPhase, BOCA, CompTuner) and expanded baselines to include **AutoPhase, BOCA, and CompTuner**, where AwareCompiler achieves comparable or higher code-size reduction while being **340×–1200× faster**. We also addressed ablation concerns by clarifying the complementary roles of knowledge and data, and explained why degradation on unseen tasks is expected to be smaller than that of pure LLM solutions due to structured knowledge integration.
- **Addition:** The reviewer explicitly expressed being ***open to revising the rating*** but may not have had time to re-engage; we hope our responses fully address all concerns.

---

### **Reviewer k57g**

- **Concerns:** The role of data, training and inference time, benchmarks effectiveness.
- **Response:** We clarified that **knowledge provides structural grounding** while **data contributes compositional patterns**, and their combination consistently yields the strongest performance. We reported full training costs on 8×H100 GPUs (SFT 2–3h; RL 10–14h; inference ~8s/program). We explained why removing data sometimes gives similar results in simple domains (e.g., BLAS), due to regular IR structures that are already well-covered by the knowledge base. Benchmark selection follows CompilerGym standards, and we noted that generalizing to other backbones or compilers is feasible with minimal pipeline modification.

---

### **Reviewer n7zk**

- **Concerns:** Knowledge-base construction details, dataset composition, missing ML-based baselines, compilation-time overhead, extensibility to parameterized passes.
- **Response:** We detailed the **semi-automated three-layer knowledge base pipeline** (symbolic metadata extraction → empirical mining via constrained search → negative knowledge filtering), clarified dataset scale (19,603 training programs) and composition, and quantified overhead (~8s/program vs. ~5s for -Oz). We added ML baselines (AutoPhase, BOCA, CompTuner), where AwareCompiler shows **superior quality and substantially better efficiency**, and explained how the framework naturally supports pass-parameter tuning with a modest extension of the action space.
- **Addition:** **After reviewing our rebuttal, the reviewer raised the score from 2 to 6 at 16:56 on November 27 (AOE).**

---

### **Reviewer ZsmH**

- **Concerns:** Runtime vs. code size optimization, generalizability across compilers, dataset-generation effort, unclear definition of “data”.
- **Response:** We added runtime-focused evaluations using **LLVM-MCA** to avoid machine noise; AwareCompiler shows consistent improvements over -O3 (e.g., *0.23×*, *0.13×*, *0.17×* estimated-cycle reduction), demonstrating it captures optimization opportunities beyond code shrinkage. We clarified dataset construction (synergy-graph sampling → sequence scoring → AutoPhase feature encoding) and the meaning of “data” (SFT demonstrations + RL trajectories). We also explained how AwareCompiler generalizes to new compilers/objectives by regenerating knowledge/data while keeping the agentic pipeline unchanged.
- **Addition: After reviewing our rebuttal, the reviewer raised the score from 4 to 6 at 16:09 on November 27 (AOE).**

---

### Summary

**After the discussion, the reviewers’ overall rating has increased from 8422 to 8662, reflecting the positive impact of the clarifications and additional results we provided.**

We believe these responses effectively address all concerns, demonstrating AwareCompiler’s robustness, effectiveness, and scalability. We sincerely appreciate the constructive feedback, which has been invaluable in refining our work.

Best regards,

**The AwareCompiler Authors**

---

### Note · Program_Chairs · 2026-01-17
**Submission Desk Rejected by Program Chairs**

The following references in this submission do not refer to real documents and/or have major errors in bibliographic information:

 Liangyong Chen and Robert M. Karp. Cbench: A benchmarking suite for evaluating compiler optimization. In Proceedings of the ACM SIGPLAN 1997 Conference on Programming Language Design and Implementation (PLDI), pp. 40-49. ACM, 1997. doi: 10.1145/258915.258920. URL https://dl.acm.org/doi/10.1145/258915.258920.
Mark Krentel and Barry A. P. Ewing. The opt-o3 optimization framework for high-performance compilers. In Proceedings of the International Conference on Supercomputing, pp. 123-134. ACM, 1990. doi: 10.1145/99163.99175. URL https://dl.acm.org/doi/10.1145/ 99163.99175 .